# Dual signaling via interferon and DNA damage response elicits entrapment by giant PML nuclear bodies

**Myriam Scherer[1], Clarissa Read[1,2], Gregor Neusser[3], Christine Kranz[3], Anna K Kuderna[1], Regina Müller[4], Florian Full[4], Sonja Wörz[1], Anna Reichel[1], Eva-Maria Schilling[1], Paul Walther[2], Thomas Stamminger[1]***

[1]Institute of Virology, Ulm University Medical Center, Ulm, Germany; [2]Central Facility for Electron Microscopy, Ulm University, Ulm, Germany; [3]Institute of Analytical and Bioanalytical Chemistry, Ulm University, Ulm, Germany; [4]Institute of Clinical and Molecular Virology, Friedrich Alexander Universität Erlangen-Nürnberg, Erlangen, Germany

*For correspondence:
thomas.stamminger@uni-ulm.de

Competing interest: The authors declare that no competing interests exist.

**Abstract** PML nuclear bodies (PML-NBs) are dynamic interchromosomal macromolecular complexes implicated in epigenetic regulation as well as antiviral defense. During herpesvirus infection, PML-NBs induce epigenetic silencing of viral genomes, however, this defense is antagonized by viral regulatory proteins such as IE1 of human cytomegalovirus (HCMV). Here, we show that PML-NBs undergo a drastic rearrangement into highly enlarged PML cages upon infection with IE1-deficient HCMV. Importantly, our results demonstrate that dual signaling by interferon and DNA damage response is required to elicit giant PML-NBs. DNA labeling revealed that invading HCMV genomes are entrapped inside PML-NBs and remain stably associated with PML cages in a transcriptionally repressed state. Intriguingly, by correlative light and transmission electron microscopy (EM), we observed that PML cages also entrap newly assembled viral capsids demonstrating a second defense layer in cells with incomplete first-line response. Further characterization by 3D EM showed that hundreds of viral capsids are tightly packed into several layers of fibrous PML. Overall, our data indicate that giant PML-NBs arise via combined interferon and DNA damage signaling which triggers entrapment of both nucleic acids and proteinaceous components. This represents a multilayered defense strategy to act in a cytoprotective manner and to combat viral infections.

## Editor's evaluation

This work adds to our understanding of cellular structures called promyelocytic leukaemia nuclear bodies (PML-NBs) in different contexts. The authors show that PML, the principal scaffolding protein of PML-NBs, forms a variety of different structures in response to viral infection, immune stimulation, and DNA damage. Further, they identify PML to restrict the replication of human cytomegalovirus (HCMV) at multiple stages of infection through the formation of alternate PML-scaffold assemblies, which gives important new insights into the interaction between this opportunistic viral pathogen and its host.

## Introduction

In order to establish a successful infection, viruses have to overcome intrinsic, innate, and adaptive host defenses which act in a cooperative manner to block viral replication and spread. The first line of intracellular defense is provided by intrinsic immune effectors that, in contrast to classical innate

responses like the interferon system, are constitutively expressed and do not require pathogen-induced activation. Studies over the last 25 years have identified promyelocytic leukemia protein nuclear bodies (PML-NBs), also known as nuclear domain 10, as key mediators of intrinsic immunity against viruses from different families (*Tavalai and Stamminger, 2008*; *Everett and Chelbi-Alix, 2007*). These highly dynamic protein complexes are located in the interchromosomal space of the cell nucleus and appear as discrete foci with a size of 0.2–1.0 µm in diameter and a number of 1–30 PML-NBs per nucleus, depending on cell type and condition (*Bernardi and Pandolfi, 2007*). PML, the structure-defining component of PML-NBs, belongs to the tripartite motif (TRIM) protein family and is expressed in at least seven isoforms whose common N-terminal region consists of a RING domain, one or two B-boxes, and a coiled-coil (CC) domain. All PML isoforms are subject to covalent modification with small ubiquitin-like modifier (SUMO) proteins, which enables the recruitment of further components and is therefore essential for PML-NB biogenesis. Due to the high number of permanently or transiently recruited proteins, PML-NBs have been implicated in a variety of cellular processes including transcriptional regulation, control of apoptosis and cellular senescence as well as DNA damage response. Furthermore, the early observation that PML-NBs target invading genomes of several DNA viruses, including the herpesviruses herpes simplex virus 1 (HSV-1) and human cytomegalovirus (HCMV), papillomaviruses and adenoviruses, raised the concept that PML-NBs act as sites for deposition of viral DNA (*Ishov and Maul, 1996*; *Ishov et al., 1997*; *Day et al., 2004*). More recent research on HSV-1 has confirmed the nuclear association with parental viral genomes and has shown that HSV-1 DNA is enveloped by PML-NBs, which thereby contribute to the control of latency in infected neurons and to intrinsic restriction of lytic HSV-1 infection (*Catez et al., 2012*; *Everett et al., 2007*; *Alandijany et al., 2018*). In contrast, a study on the interplay of PML-NBs with adenoviral genomes has found a viral DNA replication factor but not genome complexes colocalizing with PML-NBs, thus arguing against a general role as deposition site for viral genomes (*Komatsu et al., 2016*).

Characterization of the intrinsic immune function of PML-NBs during herpesvirus infection has identified the major components PML, Sp100, Daxx, and ATRX as independent restriction factors that induce epigenetic silencing of viral DNA by recruiting chromatin-modifying enzymes (*Tavalai and Stamminger, 2009*). This restrictive activity enables PML-NBs to block one of the first steps in the herpesviral life cycle. However, it is saturable and can be overcome by high doses of virus. A different antiviral mechanism, acting on a later stage of infection, has been shown to affect the herpesvirus varicella-zoster virus (VZV). During VZV infection, PML-NBs target and enclose viral nucleocapsids, mediated through a specific interaction of PML isoform IV with the ORF23 capsid protein (*Reichelt et al., 2011*; *Reichelt et al., 2012*). In addition to their role in intrinsic immunity, accumulating evidence implicates PML-NBs in the innate immune defense. An interplay between PML-NBs and innate immunity has been discovered with the observation that interferon (IFN) treatment induces the expression of specific PML-NB factors, such as PML and Sp100, and leads to an increased size and number of foci (*Chelbi-Alix et al., 1995*; *Lavau et al., 1995*). In line with this, PML-NBs participate in the establishment of an IFN-induced antiviral state and depletion of PML reduces the capacity of IFNs to protect from viral infections (*Regad et al., 2001*; *Chee et al., 2003*). Recent evidence has found that PML itself acts as a co-regulatory factor for the induction of IFN-stimulated genes, suggesting an even closer cross talk between intrinsic and innate immune mechanisms (*Ulbricht et al., 2012*; *Kim and Ahn, 2015*; *El Asmi et al., 2014*; *Scherer et al., 2016*).

In light of the broad antiviral activity of PML-NBs, it is not surprising that viruses encode antagonistic effector proteins that employ diverse strategies to inactivate single PML-NB proteins or disrupt the integrity of the whole structure. HCMV, a ubiquitous beta-herpesvirus causing serious disease in immunocompromised individuals, encodes at least two effector proteins that act in a sequential manner to efficiently antagonize PML-NB-based repression. Upon infection, the tegument-delivered protein pp71 is imported into the nucleus where it leads to dissociation of ATRX from PML-NBs, followed by proteasomal degradation of Daxx (*Lukashchuk et al., 2008*; *Saffert and Kalejta, 2006*). As this facilitates initiation of viral immediate-early gene expression, the immediate-early protein 1 (IE1) can be expressed and induces a complete dispersal of PML-NBs within the first hours of infection. Mechanistic studies have revealed that IE1 directly interacts with PML through its all alpha-helical core domain and blocks de novo SUMOylation of PML, thus disrupting PML-NB integrity (*Schilling et al., 2017*). While the PML-antagonistic protein ICP0 of HSV-1 has been shown to induce a widespread degradation of SUMO-modified proteins, IE1 uses a more specific, but yet not fully elucidated

mechanism to inhibit PML SUMOylation and thereby promote viral replication (*Boutell et al., 2011*). Due to such rapid and effective countermeasures, recombinant viruses that lack antagonistic proteins provide a valuable tool to study antiviral activities of cellular restriction factors, which would otherwise not be detectable.

Here, we report an interferon- and DNA damage signaling-induced formation of huge PML spheres, referred to as PML cages, which occurs during infection with IE1-deleted HCMV. Visualization of viral DNA with clickable fluorescent azides revealed that input HCMV genomes are entrapped by PML-NBs and remain stably encased by PML cages, leading to repression of viral gene expression as a first layer of antiviral defense. Moreover, we identify a second layer of PML-based protection for cells escaping the gene silencing-driven defense: we demonstrate that PML cages entrap newly assembled HCMV capsids in late infected cells by using correlative light and transmission electron microcopy (CLEM). 3D reconstruction of PML cages after focused ion beam-scanning electron microscopy (FIB-SEM) tomography illustrates hundreds of HCMV capsids sequestered by fibrous PML structures. Overall, these data indicate a dual, PML-based inhibition of HCMV infection and suggest entrapment of viral material as a general restriction mechanism used by PML-NBs.

## Results

### PML forms large, spherical structures after infection with IE1-deleted HCMV

PML-NBs are known to associate with HCMV genomes that enter the nucleus in order to silence viral gene expression. The antiviral structures, however, are destroyed by the immediate-early protein IE1 within the first hours of infection. To characterize the repressive effects of PML-NBs further, we analyzed their role during HCMV infection in absence of the antagonistic activity of IE1. To this end, primary human foreskin fibroblast (HFF) cells were infected with wild-type HCMV, strain AD169, or previously described recombinant cytomegaloviruses harboring either a deletion of IE1 (AD169ΔIE1) or a leucine-to-proline mutation at position 174 of IE1 that affects its structural integrity and abolishes its interaction with PML (AD169/IE1-L174P) (*Scherer et al., 2016*). In subsequent immunofluorescence analysis we observed that PML, which usually shows a diffuse, nuclear distribution in HCMV-infected cells (*Figure 1a*, panel 1), localizes to unusually large, ring-like structures after infection with recombinant cytomegaloviruses (*Figure 1a*, panel 2 and 3; *Figure 1b*). Orthogonal views and maximum intensity projection of confocal z-series images suggested that these structures are in fact spherical with PML being present at the outer layer (*Figure 1c and d*; *Video 1*). Since deletion and mutation of IE1 resulted in the same reorganization of PML-NBs during infection, both recombinant viruses were utilized for the following experiments and exemplary results are shown. To further investigate the formation of PML spheres, we performed time-course analysis in cells infected with IE1-defective HCMV and examined the subcellular localization of PML and UL44, which is a marker for viral replication centers. As illustrated in *Figure 1e*, PML-NBs were slightly enlarged at 8 hr post-infection (hpi), when compared to non-infected cells. During progression of infection, several PML foci developed into ring-like structures that were often juxtaposed to viral replication centers (*Figure 1e*, panel 4 and 5). In summary, these data show that PML-NBs, when not disrupted by IE1, undergo a drastic redistribution during HCMV infection. These newly formed structures will be referred to as PML cages, since they resemble enlarged PML-NBs that have been observed in patients with immunodeficiency, centromeric instability and facial dysmorphy (ICF) syndrome or in varicella zoster virus-infected cells and have been shown to encase cellular or viral components (*Reichelt et al., 2011*; *Luciani et al., 2006*).

### PML, but no other PML-NB component, is required for formation of PML cages

Next, we set out to characterize the architecture of PML cages. For this purpose, HFF were infected with IE1-deficient HCMV, followed by co-staining of PML and proteins that are known to permanently reside at PML-NBs. Except for ATRX (*Figure 2a*, panel 3), all main NB components, namely Sp100, Daxx, SUMO-1, and SUMO-2/3, were detected at the rim of PML cages indicating a similar composition to that of PML-NBs (*Figure 2a*, panels 1, 2, 4, and 5). This was further supported by the finding that PML is required for induction of PML cages, since other PML-NB proteins, like Sp100 or Daxx, did not localize to ring-like structures in PML-depleted HFF (*Figure 2b*, panel 1 and 2).

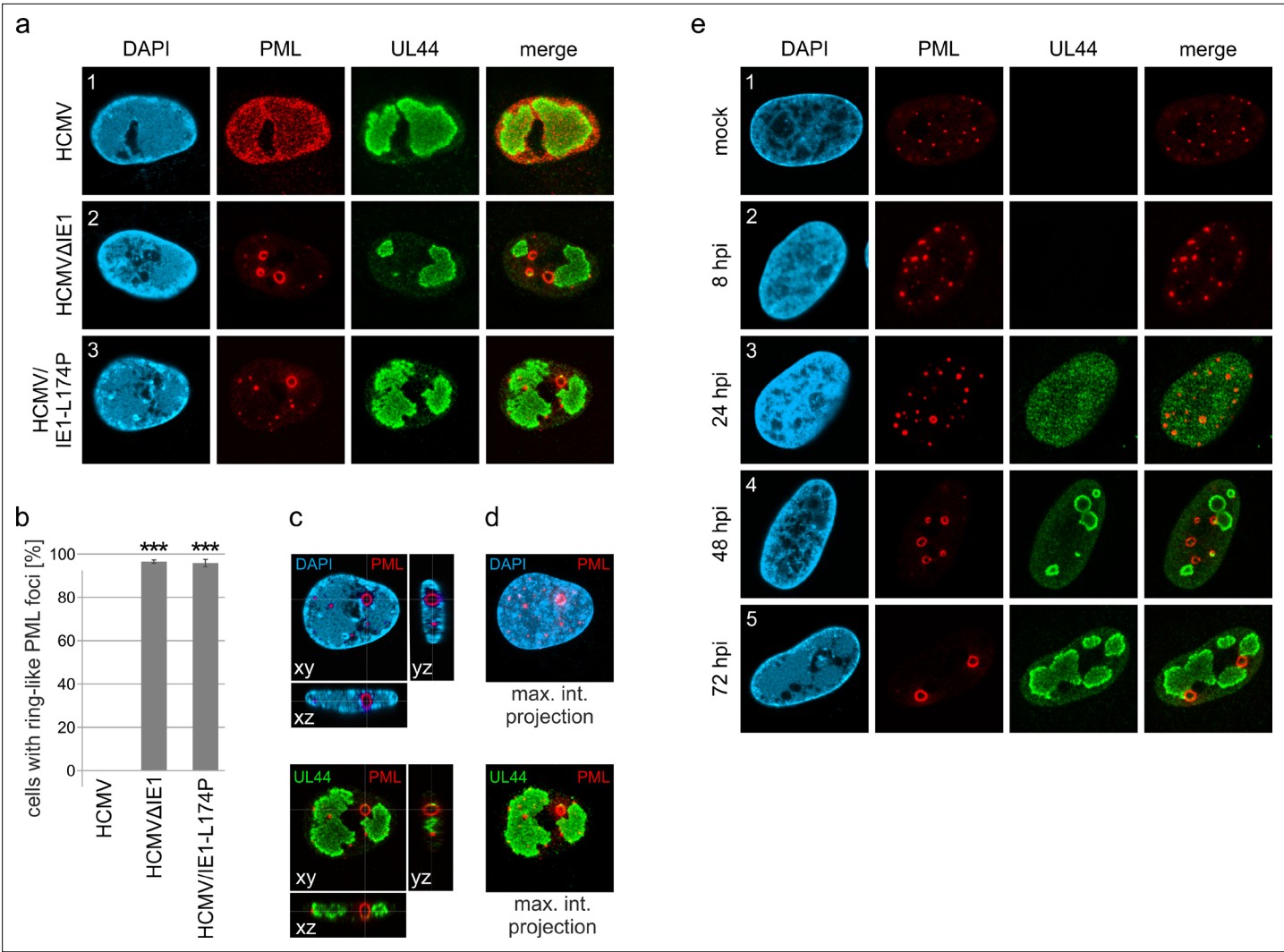

**Figure 1.** Formation of PML cages during infection with IE1-deficient HCMV. (**a–e**) HFF were infected with HCMV strain AD169, IE1-deleted AD169 (HCMVΔIE1) or AD169 encoding IE1 mutant L174P (HCMV/IE1-L174P) at a MOI of 5 IEU/cell. Cells were harvested at 72 hpi for immunofluorescence staining of endogenous PML and UL44 (**a**). Infected cells with ring-like PML structures were quantified in a population of >900 cells derived from three independent experiments (**b**). Mean values ± SD are shown and asterisks indicate significant differences; ***, p < 0.001 (see also *Figure 1—source data 1*). Orthogonal projections (**c**) and maximum intensity projection (**d**) from confocal z-series images of an HCMV/IE1-L174P-infected cell are shown. (**e**) HFF were infected with HCMV/IE1-L174P (MOI 5 IEU/cell) or not infected (mock) and harvested at indicated times for immunofluorescence analysis of endogenous PML and UL44 localization. Cell nuclei were stained with DAPI.

The online version of this article includes the following source data for figure 1:

**Source data 1.** Numerical data that are represented as a graph in *Figure 1b*.

Knockdown of Sp100, Daxx or ATRX, in contrast, did not abolish formation of PML cages (*Figure 2b*, panel 3–5). To investigate whether PML cages are built by a specific isoform, PML-depleted fibroblasts were subjected to lentiviral transduction in order to reintroduce individual FLAG-tagged PML isoforms. While overexpression of PML isoforms resulted in unspecific aggregates in non-infected cells (*Figure 2c*), infection with IE1-deficient HCMV induced a re-organization of all PML isoforms into ring-shaped structures (*Figure 2d*). We conclude that PML cages are not formed by isoform-specific interactions but require the common N-terminal domain, which contains the TRIM motif and SUMO sites mediating PML oligomerization and binding partner recruitment, respectively.

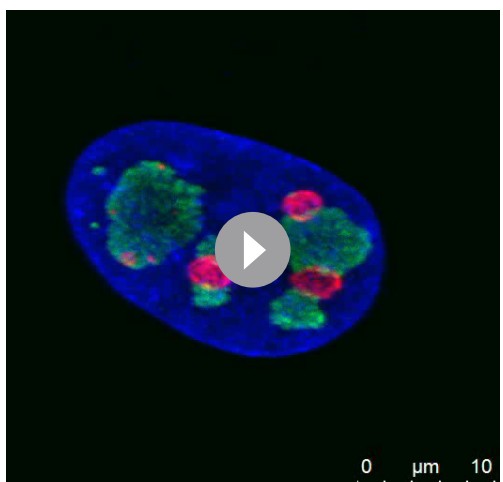

**Video 1.** Formation of PML cages after infection with HCMV-IE1-L174P. 3D reconstruction of a cell nucleus after infection with HCMV-IE1-L174P and immunofluorescence staining as described in Figure 1a. The cell nucleus is shown in blue, PML cages in red, and viral replication centers in green.
https://elifesciences.org/articles/73006/figures#video1

## Formation of PML cages is MOI-dependent but does not require viral DNA replication

IE1-deleted HCMV has been shown to grow in a multiplicity of infection (MOI)-dependent manner, as efficient early/late gene expression and DNA replication can take place only after infection with high virus doses (*Scherer et al., 2016*; *Greaves and Mocarski, 1998*). In order to evaluate whether formation of PML cages is also induced in a MOI-dependent manner, we performed a series of infection experiments with increasing amounts of IE1-deficient HCMV (MOI of 0.1–10 IEU/cell). Immunofluorescence analysis at 72 hpi revealed that size and signal intensity of PML structures increase at higher MOIs (*Figure 3a*). As individual cells showed considerable variation in number and size of PML foci, we performed ImageJ-based quantification of PML foci and divided them into three groups containing normal sized (perimeter <3 μm), enlarged (perimeter 3–10 μm), or highly enlarged PML-NBs (perimeter >10 μm). As shown in *Figure 3b* (MOI 0.1), the total number of PML foci per cell nucleus was increased after low multiplicity infection, resembling effects that were observed in interferon-treated cells (*Chelbi-Alix et al., 1995*; *Lavau et al., 1995*). Under higher MOI conditions, however, the overall number of PML foci decreased again (*Figure 3b*), while a higher percentage of unusually enlarged PML-NBs was detected (*Figure 3c*). Since strongly enlarged PML foci were found particularly after infection with MOIs greater than 1, which allow lytic replication and viral early/late gene expression (*Figure 3d*), the question arose whether viral DNA replication is required for their formation. To analyze this, HFF were treated with viral DNA polymerase inhibitor foscarnet (PFA) or were mock treated at 1.5 hr after infection with IE1-deleted HCMV, before being subjected to immunofluorescence staining of PML and UL44 as a marker for replication centers (*Figure 3e*). Subsequent quantification of PML foci size revealed a slight but significant reduction of highly enlarged PML structures in the presence of PFA (*Figure 3f*). Overall, however, no abrogation of PML cages was observed (*Figure 3e, f*) suggesting that formation of these structures depends on the amount of input virus but does not require viral DNA replication.

## Interferon and DNA damage signaling act in a cooperative manner to induce PML cages

Previous studies of our and other groups have shown that IE1 blocks IFN signaling during HCMV infection by affecting both STAT proteins and PML, which has been identified as positive regulator of IFN-induced gene expression (*Kim and Ahn, 2015*; *Scherer et al., 2016*; *Paulus et al., 2006*). Since expression of several PML-NB proteins, including Sp100 and PML itself, can be upregulated by IFN treatment, we hypothesized that PML cages may arise due to IFN-mediated increase of PML-NB protein levels during infection with IE1-deleted virus. To assess the role of IFN signaling for formation of PML cages, HFF were incubated with IE1-defective HCMV alone, or treated with an anti-IFNα/β receptor antibody (mAb-IFNAR) prior to virus inoculation. Western blot analysis at 72 hpi revealed considerably higher abundances of all PML and Sp100 variants in comparison to mock infected cells (*Figure 4a*, lane 1 and 2). Block of IFN signaling by mAb-IFNAR-antibody treatment, which was verified by lack of STAT2 phosphorylation, largely reverted this effect and facilitated virus infection, as reflected by higher levels of viral early and late proteins (*Figure 4a*, lane 3). These data correlated with the observation that mAb-IFNAR-antibody treatment resulted in a clear reduction of enlarged PML structures in infected cells, as assessed by immunofluorescence staining and quantification of PML foci size (*Figure 4b and c*). Having observed that IFN signaling plays an important role for the formation

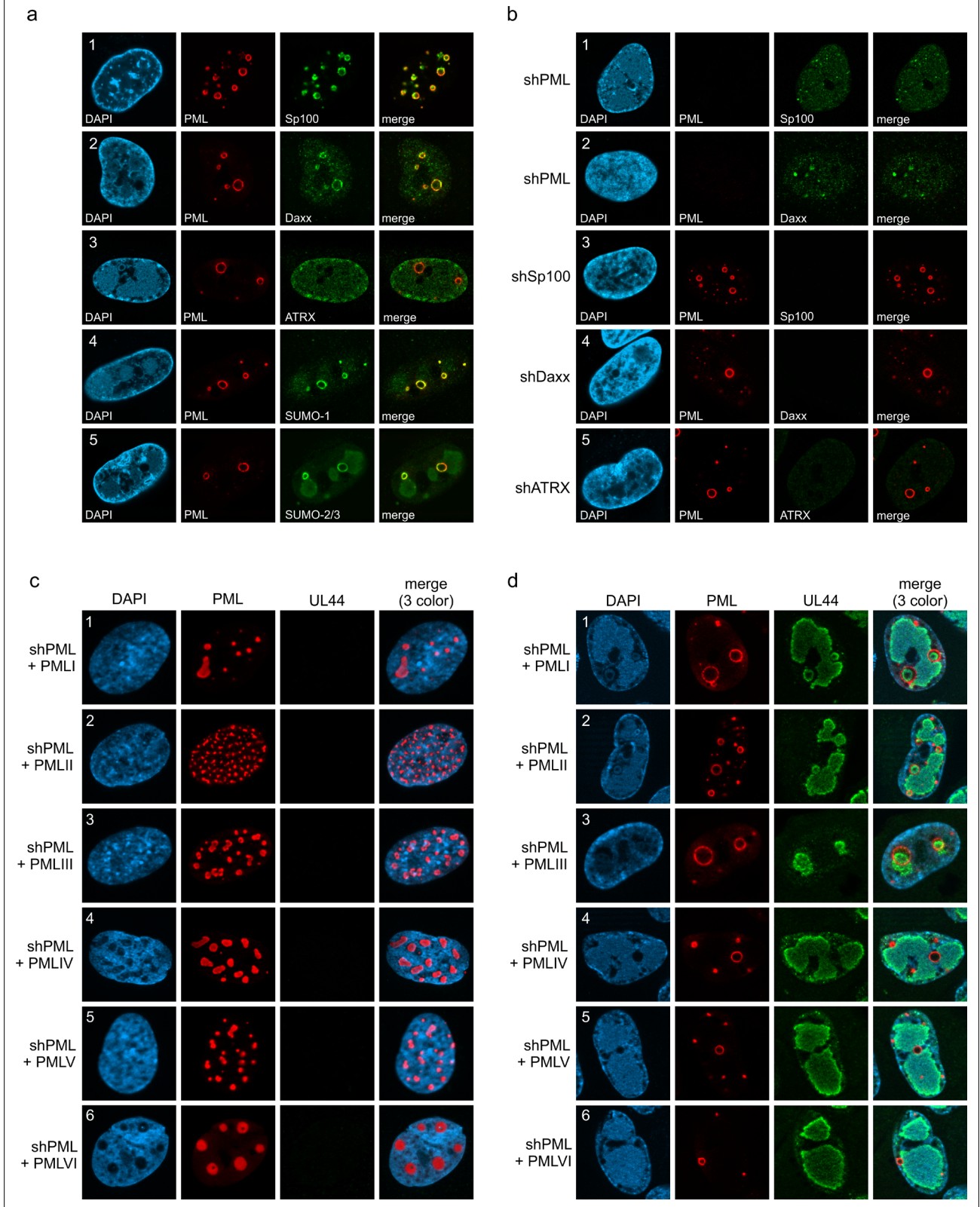

**Figure 2.** Protein composition of PML cages. (**a**) Recruitment of nuclear body proteins to PML cages. HFF were infected with HCMV/IE1-L174P, based on strain AD169, at a MOI of 5 IEU/cell and harvested at 72 hpi for immunofluorescence staining of PML together with NB components Sp100, Daxx, ATRX, SUMO-1, and SUMO-2 as indicated. (**b**) PML as key organizer of PML cages. HFF depleted for PML (shPML), Sp100 (shSp100), Daxx (shDaxx) or ATRX (shATRX) were infected with HCMVΔIE1, based on strain AD169, at a MOI of 5 IEU/cell and harvested at 72 hpi for staining of PML-NB proteins.

*Figure 2 continued on next page*

*Figure 2 continued*

(**c, d**) PML isoform-independent formation of PML cages. Flag-tagged PML isoforms I-VI were reintroduced into PML-knockdown HFF by lentiviral transduction. Newly generated cells were non-infected (**c**) or infected with HCMVΔIE1 at MOI 10 (IEU/cell) for 72 h (**d**) and were stained with antibodies directed against FLAG and PML. DAPI staining was performed to visualize cell nuclei.

of PML cages during infection, we next analyzed whether IFN treatment alone can induce such effects in HFF cells. Treatment of HFF with IFN-β for 3 d or even 6 d resulted in a strong upregulation of all PML and Sp100 variants (*Figure 4d*), similar as in cells infected with IE1-deficient HCMV (*Figure 4a*). In immunofluorescence analysis, we detected an enhanced signal intensity and number of PML-NBs after IFN treatment. However, only few ring-like structures and no strongly enlarged PML cages were observed (*Figure 4e and f*), suggesting that IFN-based upregulation of PML-NB proteins is not sufficient to induce these structures but that a further stimulus is required which is provided by the virus. Since rearrangements of PML-NBs have recently been linked to cellular DNA damage, we speculated that DNA damage signaling, induced by incoming HCMV DNA, is required in addition to IFN signaling (*Imrichova et al., 2019*). To test this, HFF were treated with IFN or were mock treated, and DNA damage was induced by addition of topoisomerase I inhibitor camptothecin (CPT) or topoisomerase II inhibitor doxorubicin (Doxo). As illustrated in *Figure 4g*, different PML structures were observed, including small and large PML cages as well as structures termed caps and forks. Intriguingly, while CPT treatment alone for 2d or 4d resulted in small PML structures (*Figure 4g*; *Figure 4—figure supplement 1a*, panels 3 and 5) in few cells, co-treatment with IFN evoked also large versions of these structures (*Figure 4g*; *Figure 4—figure supplement 1a*, panels 4 and 6) with PML cages being present in the majority of cells (*Figure 4h*). HFF cells treated with Doxo displayed all types of PML structures without IFN addition (*Figure 4h*; *Figure 4—figure supplement 1a*, panel 7). Co-treatment of HFF with Doxo and IFN, however, resulted in a rearrangement of PML-NBs to large cages in >80% of the cells (*Figure 4h*; *Figure 4—figure supplement 1a*, panel 8), similar to the effect observed in HCMVΔIE1-infected cells. Induction of DNA damage in this experimental setup was confirmed by immunostaining of the DNA damage marker γH2AX. While γH2AX foci formed shortly after treatment with CPT and Doxo (*Figure 4—figure supplement 1b*), large PML structures were observed only several days after DNA damage induction (*Figure 4—figure supplement 1c*), suggesting that the rearrangement of PML may be a response to long-term DNA damage. In order to investigate whether PML cages form in the proximity of DNA damage sites, HFF cells were treated with Doxo and IFN followed by immunostaining of PML and γH2AX (*Figure 4i*). Quantification of γH2AX-positive PML cages revealed that γH2AX can be detected at approximately 80% of all PML cages indicating a strong association between DNA damage sites and enlarged PML structures (*Figure 4j*). To corroborate this finding, we analyzed whether inhibition of DNA damage signaling affects the formation of PML cages in the context of HCMV infection. For this, HFF cells infected at either low or high MOI with HCMVhIE1, were treated with ATM inhibitor Ku-55933. Quantification of PML foci size at 72 hpi revealed a significant reduction of large PML cages in ATM inhibitor treated cells (*Figure 4k, l*). Importantly, this was also observed under low MOI conditions, where HCMVΔIE1 is not able to initiate viral DNA replication, suggesting that incoming viral genomes are sufficient to trigger DNA damage signaling (*Figure 4l*). In summary, this evidence implies that IFN and DNA damage signaling act in a cooperative manner to drive the formation of PML cages.

## PML-NBs entrap HCMV genomes upon entry into the nucleus

As viral DNA has been shown to colocalize with PML bodies upon entry into the nucleus, we next examined whether a direct association with HCMV genomes triggers PML cages (*Ishov et al., 1997*). To visualize HCMV DNA, we made use of alkyne-modified nucleosides that, after incorporation into viral DNA, can be detected with fluorescent azides by click chemistry in combination with antibody staining. Comparison of the incorporation efficiency of EdC (ethynyl-deoxycytidine), EdU (ethynyl-deoxyuridine), and F-ara-Edu (deoxy-fluoro-ethynyluridine) at different concentrations demonstrated that EdC was slightly more sensitive for detection of HCMV replication centers than EdU and significantly more efficient than F-ara-Edu (*Figure 5—figure supplement 1*). The lowest dose of 0.1 μM, which was sufficient to stain viral replication centers, was chosen to generate labeled virus stocks as it had a smaller effect on virus yields compared to higher doses (*Figure 5—figure supplement 2a*) and did neither affect viral entry into fibroblasts nor the onset of HCMV gene expression (*Figure 5—figure*

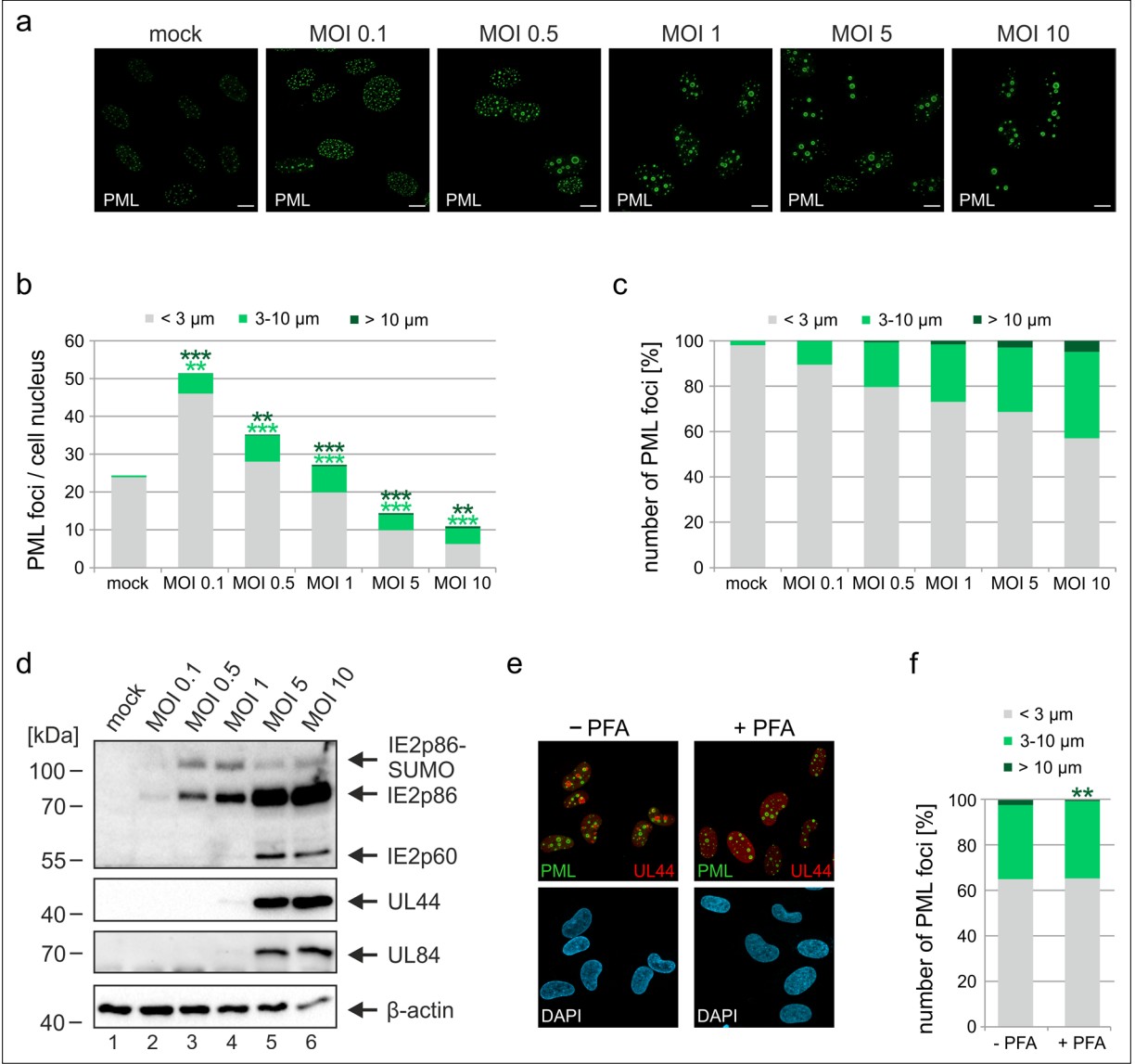

**Figure 3.** Impact of virus dose and viral DNA replication on formation of PML cages. (**a–d**) MOI-dependent induction of PML cages. HFF were infected with HCMVΔIE1, based on strain AD169, at MOIs ranging from 0.1 to 10 IEU/cell or not infected (mock) and were harvested after 72 hr. Immunofluorescence analysis was performed to analyze number and size of PML foci; scale bar, 10 µm (**a**). ImageJ-based quantification of the PML foci number and size, determined as perimeter, was performed using maximum intensity projections of confocal z-series images of >150 cells derived from three independent experiments. At MOIs < 1, only infected cells were included in the analysis, which were identified by co-staining of immediate-early protein 2 (IE2) (not shown). PML foci were separated into three groups of normal sized PML-NBs (perimeter <3 µm), enlarged PML foci (perimeter 3–10 µm), and highly enlarged PML cages (perimeter >10 µm). Shown are the mean values of PML foci numbers per cell nucleus (**b**) and the percentage total PML-NBs (**c**). Green and dark green asterisks indicate statistically significant higher numbers of enlarged PML and highly enlarged PML foci, respectively, in infected as compared to not infected cells. (**d**) Western blot detection of viral immediate-early (IE2p86), early (UL44, UL84), and late (IE2p60) proteins. Staining of β-actin was included as internal control. Full blots are shown in *Figure 3—source data 2*. (**e, f**) Formation of PML cages in absence of viral DNA replication. HFF were infected with HCMVΔIE1 at a MOI of 5 IEU/cell and were treated with 250 µM PFA in parallel with virus inoculation or were left untreated. At 72 hpi, cells were fixed for immunofluorescence staining of PML, UL44 as marker for viral replication centers, and of cell nuclei with DAPI (**e**), followed by quantification of PML foci size in >160 cells derived from three independent experiments as described above. Significant differences in the occurrence of highly enlarged PML cages are indicated by asterisks. **, p < 0.01; ***, p < 0.001. PFA, phosphonoformic acid. See also *Figure 3—source data 1*.

The online version of this article includes the following source data for figure 3:

**Source data 1.** Numerical data that are represented as graphs in *Figure 3b, c and f*.

**Source data 2.** Uncropped western blot images for *Figure 3d*.

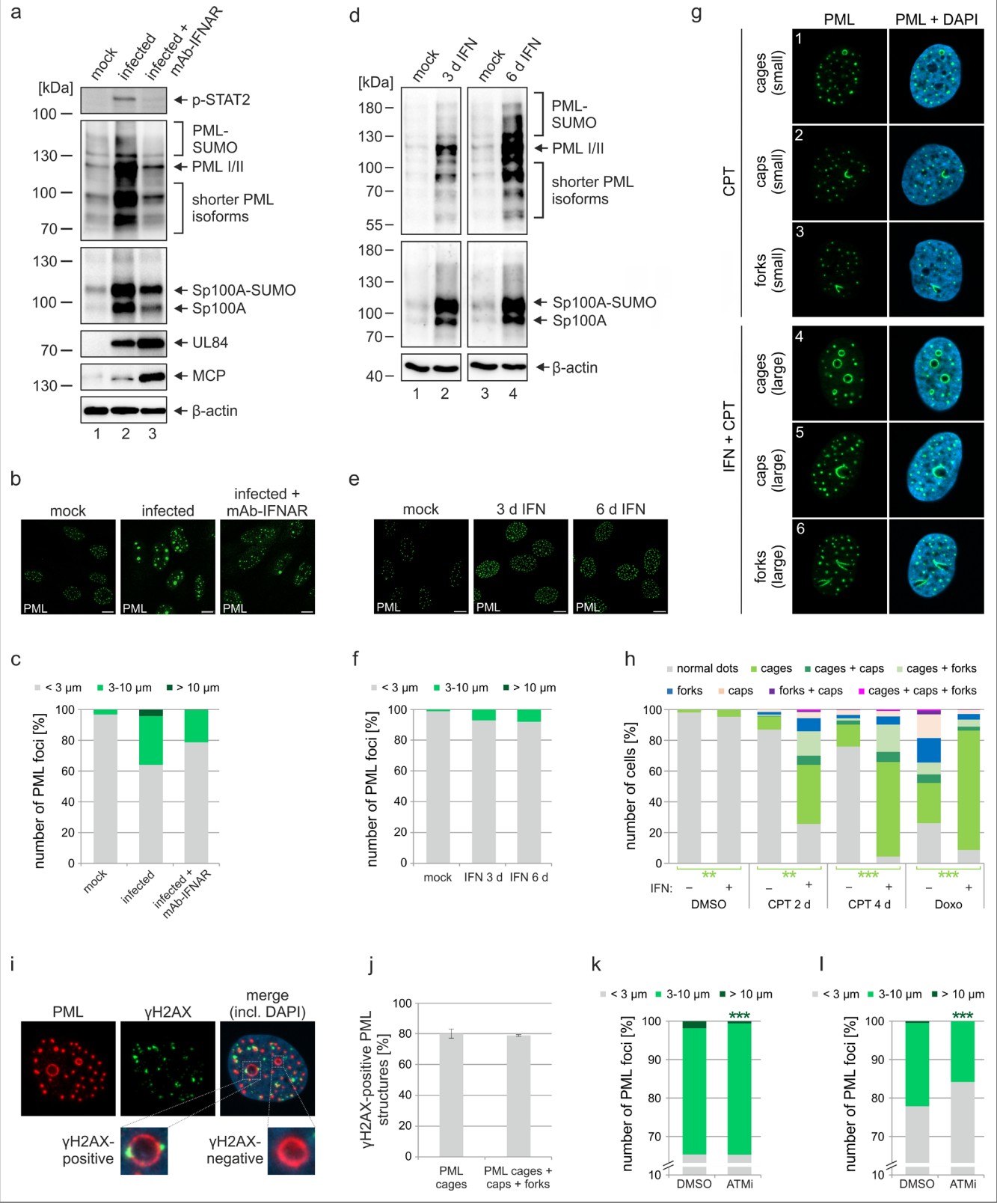

**Figure 4.** Role of IFN and DNA damage signaling for induction of PML cages. (**a–c**) Reduced formation of PML cages in infected cells upon inhibition of the IFN pathway. HFF were infected with HCMV/IE1-L174P, based on strain AD169, at a MOI of 5 IEU/ml or were not infected (mock). Monoclonal anti-IFNα/β receptor antibody (mAb-IFNAR) was added 25 min prior to infection at a concentration of 5 μg/ml. Cells were harvested at 72 hpi for Western blot detection using antibodies directed against phosphorylated STAT2, PML, Sp100, viral early protein UL84, viral late protein MCP, and

*Figure 4 continued on next page*

*Figure 4 continued*

β-actin as control (**a**) (see *Figure 4—source data 1* for full blots) or for immunofluorescence analysis of PML foci (**b**). Maximum intensity projections of confocal z-series images were used to quantify number and size of PML foci in ≥50 cell nuclei per sample (**c**). (**d-f**) No formation of PML cages by IFN treatment alone. HFF were treated with IFN-β (1,000 U/ml) for 3 d or 6 d or were left untreated (mock). Cells were subjected to Western blot analysis of PML, Sp100, and β-actin as loading control (**d**) (see *Figure 4—source data 1* for full blots) or were harvested for immunofluorescence staining of PML (**e**); scale bar, 10 μm. Maximum intensity projections of confocal z-series images were used to quantify number and size of PML foci in ≥160 cell nuclei derived from three independent experiments (**f**). Scale bars, 10 μm. (**g–h**) Formation of PML cages after stimulation of IFN and DNA damage signaling. (**g**) Overview of PML structures (cages, caps, and forks) forming upon induction of DNA damage in HFF cells. HFF cells were seeded at low density (30,000 cells / well in 12-well plates) and were then mock treated (panels 1–3) or treated with 1000 U/ml IFN-β for 3 days (panels 4–6), followed by addition of CPT at a final concentration of 1 μM for 2 days. Cells were stained for endogenous PML. (**h**) Quantification of PML structures in HFF cells treated with DNA damage-inducing chemicals CPT and Doxo. HFF cells were seeded at low density (30,000 cells / well in 12-well plates) and, the next day, were mock treated (-) or treated with 1000 U/ml IFN-β (+) for 3 days, followed by addition of CPT or Doxo at final concentrations of 1 μM and 0.5 μM, respectively, or DMSO as control. Cells were harvested 2 days later (DMSO, Doxo, CPT 2 d) or 4 days later (CPT 4 d) and subjected to immunofluorescence staining of PML. Cell nuclei were visualized with DAPI. Formation of PML structures (cages, forks, circles as well as combinations of these structures as indicated) was assessed in >1300 cells derived from three independent experiments. Statistically significant differences in the occurance of cells with PML circles in IFN-treated groups compared to not IFN-treated groups are indicated by asterisks. **, p < 0.01; ***, p < 0.001. (**i, j**) Association of PML structures with sites of DNA damage. HFF cells were treated with Doxo and IFN as described in (**h**), followed by immunofluorescence staining of PML and γH2AX. (**i**) The numbers of γH2AX-positive PML cages, as shown in the insets, as well as all enlarged PML structures (cages, caps, and forks) were determined in >160 cells derived from three independent experiments (**j**). (**k, l**) Reduced formation of PML cages upon inhibition of DNA damage signaling. HFF were infected with HCMVΔIE1 at a MOI of 3 IEU/cell (**k**) or a MOI of 0.5 IEU/cell (**l**) and were treated with ATM inhibitor Ku-55933 (30 μM) 10 min prior to virus inoculation, or were treated with DMSO as control. At 72 hpi, cells were fixed for immunofluorescence staining of PML followed by quantification of PML foci size in >160 cells derived from three independent experiments. Statistically significant differences in the number of highly enlarged PML cages are indicated by asterisks. ***, p < 0.001. IFN, interferon; CPT, camptothecin; Doxo, doxorubicin; ATMi, ATM inhibitor. See also *Figure 4—source data 2*.

The online version of this article includes the following source data and figure supplement(s) for figure 4:

**Source data 1.** Uncropped western blot images for *Figure 4a and b*.

**Source data 2.** Numerical data that are represented as graphs in *Figure 4c, f, h, j, k and l*.

**Figure supplement 1.** Effect of IFN and DNA damage on PML localization.

*supplement 2b*). Since EdC had a lower impact on virus growth (*Figure 5—figure supplement 2a*) and resulted in a higher labeling efficiency compared to EdU (*Figure 5—figure supplement 2c*, d), subsequent infection of HFF cells was performed with an EdC-labeled IE1-deletion virus termed HCMVΔIE1_EdC. At 8 hpi, signals of viral DNA (vDNA) were detectable inside cell nuclei (*Figure 5a*, panel 1–3), but not observed in cells infected with unlabeled virus (*Figure 5a*, panel 4). Closer investigation of vDNA localization revealed that these signals are frequently associated with slightly enlarged PML foci, which appeared to form a shell around HCMV genomes (*Figure 5a*, panels 1 and 2, arrows). These data implied an entrapment of HCMV genomes by PML-NBs that was confirmed by deconvolution and 3D reconstruction of z-series images (*Figure 5b*). Intriguingly, entrapment of HCMV genomes upon low MOI infection was found to be highly efficient when only one genome was present in the cell nucleus but was significantly reduced with increasing number of genomes (*Figure 5a*). This was in line with a limited entrapment of viral genomes under higher MOI conditions (*Figure 5c*), which indicates saturation of PML-NB-based intrinsic defense and provides an explanation for the MOI-dependent restriction of HCMV IE gene expression (*Tavalai et al., 2006*). In accordance, nuclear entry of only one genome did not allow IE2 expression in the majority of cells, while the presence of >3 genomes always resulted in initiation of HCMV gene expression (*Figure 5d*). Notably, at higher MOIs, genomes frequently switched from a very condensed state to more irregular and expanded shapes that colocalized with the viral transactivator protein IE2 and are indicative for decompaction and transcriptional activation of HCMV genomes (compare *Figure 5a*, panel 1 with *Figure 5c*, panel 1). Analogous results were obtained for wild-type HCMV_EdC, since low MOI infection resulted in genome entrapment as long as IE1 was not expressed (*Figure 5e*, panel 1). Under high MOI conditions, viral genomes were released by IE1-based disruption of PML-NBs and had a more decondensed appearance (*Figure 5e*, panel 2). Finally, we examined whether the PML-NBs that have encased input viral genomes develop into PML cages at late times after infection. Interestingly, input genomes were still detectable at 72 hr after low MOI infection and were exclusively found inside large PML structures, independent of the number of genomes per nucleus (*Figure 5f*, panel 1). At higher MOI (MOI of 1), viral genomes again displayed a more diffuse staining pattern allowing low level IE2 expression but not formation

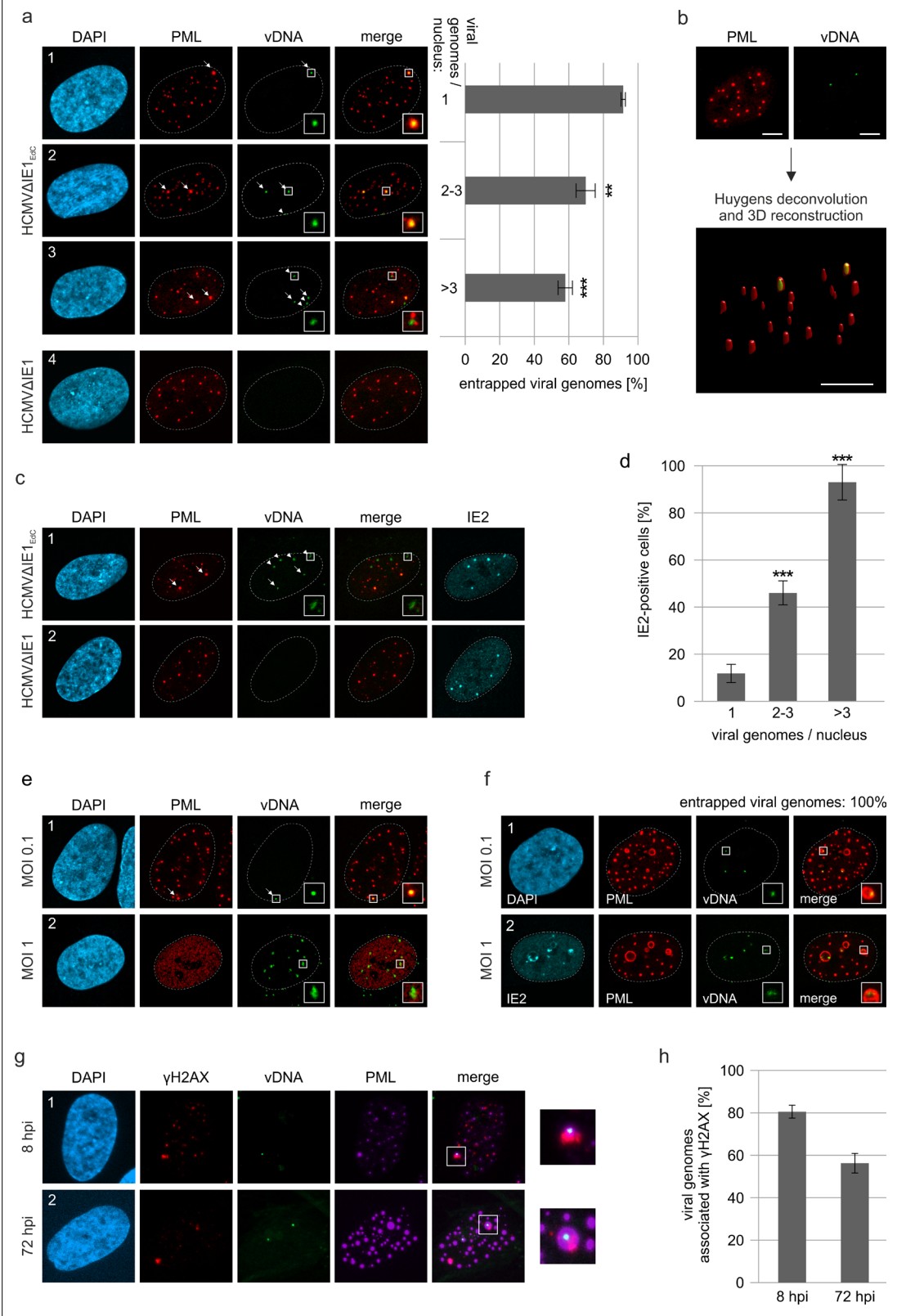

**Figure 5.** Entrapment of HCMV input genomes by PML cages. HFF were infected with EdC-labeled (HCMVΔIE1$_{EdC}$) and unlabeled HCMVΔIE1 or respective wild-type viruses (HCMV$_{EdC}$, HCMV), which are all based on strain AD169. At different times after infection, cells were fixed for antibody staining to detect IE2 and/or PML in combination with click chemistry to visualize HCMV input genomes (vDNA). (**a**) HFF were infected with HCMVΔIE1$_{EdC}$ or unlabeled HCMVΔIE1 at a MOI of 0.1 IEU/cell and stained for PML and vDNA at 8 hpi. According to the number of genomes detected

*Figure 5 continued on next page*

*Figure 5 continued*

in the nucleus, cells were divided in three groups. Colocalization of PML and vDNA (entrapment) within these groups was determined in ≥160 cells derived from three independent experiments. Genomes encased by PML are marked by arrows, not entrapped genomes by arrowheads. Cell nuclei were visualized by DAPI staining. Asterisks indicate significant differences as compared to the group of cells with 1 viral genome/nucleus (**, p-value < 0.01; ***, p-value < 0.001). (**b**) 3D reconstruction of a confocal image stack showing PML-NBs in red and HCMV genomes in green. Scale bar, 5 µm. (**c**) HFF were infected with HCMVΔIE1$_{EdC}$ or HCMVΔIE1 at a MOI of 1 IEU/cell for 8 hr, before they were fixed for immunofluorescence analysis of PML and IE2 combined with click chemistry. (**d**) HFF were infected, stained, and grouped as described in (**a**). IE2-positive cells were determined by immunofluorescence staining in ≥200 cells derived from three independent experiments and are shown as percentages of each group. Asterisks indicate significant differences as compared to the group of cells with 1 viral genome / nucleus (***, p-value < 0.001) (**e**) Infection of HFF was performed with HCMV$_{EdC}$ at a MOI of 0.1 or 1 IEU/cell as indicated. PML and vDNA were stained at 8 hpi. (**f**) HFF were infected with HCMVΔIE1$_{EdC}$ at a MOI of 0.1 or 1 IEU/cell as indicated and 72 hpi, cells were fixed for co-staining of PML, IE2, and vDNA. vDNA entrapment after infection with MOI 0.1 was determined in >190 cells derived from three independent experiments. (**g, h**) Association of entrapped vDNA with DNA damage sites. HFF were infected with HCMVΔIE1$_{EdC}$ at a MOI of 0.1 IEU/cell and were fixed at 8 hpi or 72 hpi for staining of vDNA, γH2AX, and PML as indicated (**g**). Entrapped HCMV genomes were analyzed for an association with γH2AX in >160 cells derived from three independent experiments and the percentages of γH2AX-positive structures are presented as mean values ± SD (**h**). See also *Figure 5—source data 1*.

The online version of this article includes the following source data and figure supplement(s) for figure 5:

**Source data 1.** Numerical data that are represented as graphs in *Figure 5a, d, f and h*.

**Figure supplement 1.** Detection of HCMV replication centers using alkyne-modified nucleosides.

**Figure supplement 2.** Effect of alkyne-modified nucleosides on HCMV growth and IE gene expression.

**Figure supplement 2—source data 1.** Numerical data that are represented as graphs in *Figure 5—figure supplement 2a, b and d*.

---

of viral replication compartments (*Figure 5f*, panel 2). Since vDNA signals were found in both small and large PML foci and in addition, not all of the large PML cages contained HCMV genomes, we assume that their enlargement may not be a direct consequence of the entrapment process. Finally, we wanted to know whether entrapped vDNA exhibits an association with DNA damage sites. HFF cells were infected with HCMVΔIE1$_{EdC}$ at low MOI followed by staining of vDNA, PML and the DNA damage marker γH2AX at either 8 hpi or 72 hpi (*Figure 5g*). Quantification revealed that, although a slight decrease of association was observed at late times after infection, the majority of vDNA signals could be detected in the vicinity of DNA damage sites (*Figure 5h*). Taken together, our data provide evidence that genome entrapment by PML-NBs has evolved as a mechanism to achieve efficient and persistent repression of HCMV. Reorganization from dot-like foci to PML cages, however, may not be directly linked to vDNA entrapment but seems to require an additional stimulus.

## IE1 expression disrupts PML cages to abrogate the repression of HCMV genomes and also induces dispersal of ALT-associated PML-NBs

Inhibition of HCMV gene expression by PML-NBs has been shown to involve the recruitment of chromatin-associated factors with repressive activity (*Tavalai and Stamminger, 2009*). In line with this, we detected the chromosomal DNA-binding protein HP1α (heterochromatin protein 1 alpha) at PML cages, which is involved in establishing a stable heterochromatic structure and is usually excluded from replication centers during wild-type HCMV infection (*Figure 6a*). Since Sp100 was described to interact with HP1 proteins, we analyzed the distribution of HP1α in Sp100-depleted cells after HCMVΔIE1 infection (*Seeler et al., 1998*). Indeed, no colocalization with PML cages was observed in absence of Sp100 (*Figure 6b*, panel 2), whereas knockdown of Daxx did not abolish HP1α recruitment (*Figure 6b*, panel 3). To get further evidence for an antiviral function of PML cages, we tested whether disruption of PML cages can abrogate the repression of entrapped genomes and promote lytic infection. Lentiviral expression of IE1 was used to disrupt PML cages in cells that had been infected with HCMVΔIE1 for 72 hr, resulting in a nuclear diffuse distribution of PML (*Figure 6c*, panel 1 and 2). Intriguingly, disruption of PML cages by IE1 led to an increase of intracellular viral DNA, thus demonstrating a positive effect on HCMV DNA replication (*Figure 6d*). Since the efficiency of lentiviral transduction, as assessed by quantification of IE1-expressing cells, correlated with the expression of the HCMV early gene UL44, we conclude that IE1 induces a full relieve of PML-based transcriptional repression (*Figure 6e*). To exclude that this effect was based on STAT2 binding and inhibition of IFN signaling by IE1, HCMVΔIE1-infected cells were transduced with lentiviruses expressing the IE1 core region (IE1 1–382), which contains the PML-binding domain but lacks the STAT-binding region. Consistent with published data, IE1 1–382 did not induce a complete disruption of PML structures,

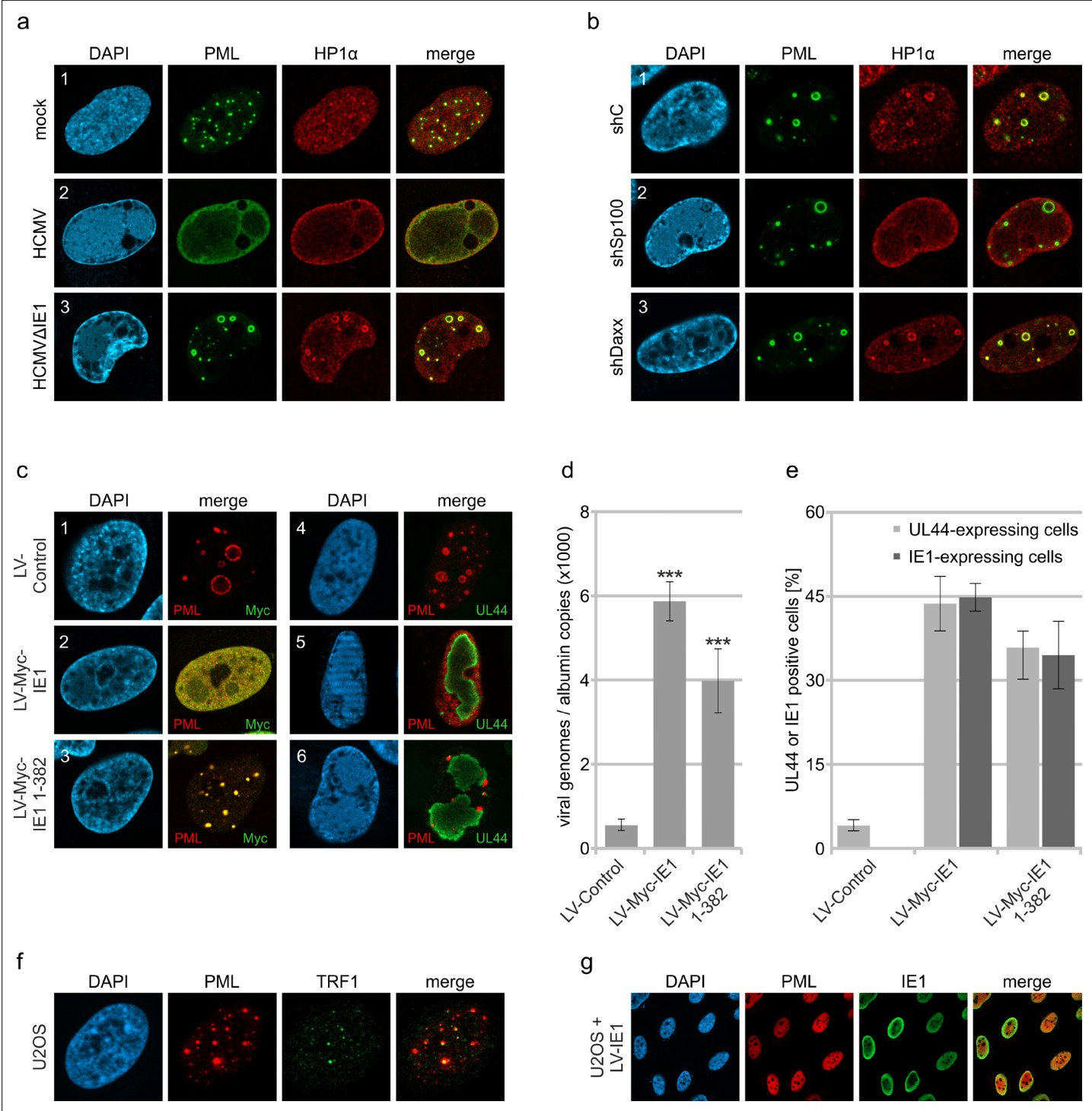

**Figure 6.** Abrogation of HCMV genome repression by disruption of PML cages. (**a**) Recruitment of heterochromatin protein HP1α to PML cages. HFF were non-infected or infected at a MOI of 5 IEU/cell with HCMV or HCMVΔIE1, based on strain AD169. After 72 hr, cells were fixed and stained with antibodies directed against PML and HP1α. (**b**) Recruitment of HP1α by Sp100. Control HFF (shC), Sp100-knockdown cells (shSp100), and Daxx-knockdown cells (shDaxx) were infected with HCMVΔIE1 (MOI of 5), followed by immunofluorescence analysis of PML and HP1α at 72 hpi. (**c**) Disruption of PML cages by lentiviral expression of IE1. HFF were infected with HCMVΔIE1 at a MOI of 1 IEU/cell for 72 hr, before they were transduced with lentiviruses encoding Myc-tagged IE1 (LV-Myc-IE1), Myc-tagged IE1 1–382 (LV-Myc-IE1 1–382) or control lentiviruses (LV-control). Four days after transduction, cells were analyzed by immunofluorescence staining of either PML and Myc-IE1 (subpanels 1–3) or PML and UL44 (subpanels 4–6). Nuclei were counterstained with DAPI. (**d, e**) Increase of viral DNA replication upon disruption of PML cages. HFF were treated as described in (**c**), followed by isolation of total DNA and quantification of viral genome copy numbers by real-time PCR (**d**) or by immunofluorescence staining of Myc-IE1 or UL44

*Figure 6 continued on next page*

*Figure 6 continued*

followed by quantification of protein expression in >500 cells per sample (**e**). Values are derived from biological quadruplicates (**d**) or triplicates (**e**) and represent mean values ± SD. Asterisks indicate statistically significant differences. ***, p < 0.001. (**f, g**) Disruption of ALT-associated PML-NBs (APBs) by IE1. (**f**) U2OS cells were stained for endogenous PML and telomere-binding protein TRF1. (**g**) U2OS cells were transduced with lentiviruses expressing IE1 followed by immunofluorescence analysis of endogenous PML and IE1. See also *Figure 6—source data 1*.

The online version of this article includes the following source data for figure 6:

**Source data 1.** Numerical data that are represented as graphs in *Figure 6d and e*.

but colocalized with PML at residual foci that have no antiviral activity (*Figure 6c*, panel 3) (*Scherer et al., 2014*). This redistribution of PML cages by IE1 1–382 also resulted in enhanced UL44 expression (*Figure 6e*) and HCMV DNA replication (*Figure 6d*) thereby corroborating the hypothesis that PML cages act as antiviral structures.

To explore the effect of IE1 on other enlarged PML structures described in literature, we made use of U2OS cells, in which the alternative lengthening of telomere (ALT) pathway is used for maintenance of telomers. In accordance with published data, we detected ALT-associated PML bodies (ABPs) in a minor percentage of cells, which have been suggested to participate in telomeric maintenance and appeared as enlarged PML-NBs colocalizing with telomere-binding protein TFR1 (*Figure 6f*; *Grobelny et al., 2000*). Upon expression of IE1 by lentiviral transduction, PML-NBs were completely dispersed in U2OS cells indicating that IE1 does not only disrupt antiviral PML cages but also other enlarged PML structures with cytoprotective function (*Figure 6g*).

## PML cages enclose newly assembled viral capsids at late stages of infection

Having observed that PML cages arise in close proximity to viral replication centers (*Figure 1*), it was tempting to speculate that these structures exert an additional antiviral activity during the late phase of infection. This was supported by the finding that major capsid protein MCP displayed a clearly altered localization in cells infected with HCMVΔIE1 in comparison to wild-type HCMV-infected cells (*Figure 7a*). Since MCP was enriched in PML cages during HCMVΔIE1 infection (*Figure 7a*, panel 2), we applied correlative light and electron microscopy (CLEM) to investigate whether viral capsid proteins or whole nucleocapsids are sequestered by PML. For this purpose, HFF cells expressing mCherry-tagged PML were established by lentiviral transduction (*Figure 7—figure supplement 1*) and infected with recombinant HCMV encoding eYFP-tagged IE2, which enabled us to allocate PML-positive and infected cells (*Wagenknecht et al., 2015*). After light microscopy (LM) imaging, cells were analyzed by transmission electron microscopy (TEM) followed by correlation of LM and TEM images. Analysis of cells with high mCherry-PML expression revealed that PML assembles to multi-layered, fibrous structures (*Figure 7b*). After infection with HCMV-IE2eYFP, PML aggregates were completely dispersed and, in accordance, no fiber-like structures were detected in EM images (*Figure 7—figure supplement 2*). While these cells showed an even distribution of viral capsids throughout the cell nuclei (*Figure 7—figure supplement 2*), we observed unusual accumulations of capsids after infection with IE1-deficient HCMV (HCMVΔIE1-IE2eYFP) (*Figure 7b*). The capsid clusters were surrounded by several layers of PML fibers and correlated with mCherry-PML signals in LM images, clearly demonstrating an entrapment of HCMV capsids by PML cages. Similar capsid accumulations were detected in non-transduced HFF after infection with IE1-deficient viruses based on strain TB40E or AD169, suggesting a virus strain-independent entrapment by endogenous PML (*Figure 7—figure supplement 3*). To characterize the capsid packing in more detail, we performed 3D imaging of mCherry-PML- and IE2-eYFP-positive cells by focused ion beam-scanning electron microscopy (FIB-SEM) with TEM-like resolution. As shown in *Figure 7d*, we analyzed three consecutive areas within a cell nucleus by FIB-SEM tomography, containing four PML cages in total (*Video 2*). Similar to the TEM analysis, PML cages appeared as spherical, fibrous structures encasing tightly packed nuclear capsids (*Figure 7d*, panels A and C) and, in some cases, additional electron-dense material likely composed of viral DNA or protein aggregates (*Figure 7d*, panel B). 3D reconstruction of a PML cage containing exclusively capsids identified 366 capsids in an inner cage volume of 0.31 μm$^3$, so that a high packing density of about 1,180 nuclear capsids/μm$^3$ can be estimated (*Figure 7e*; *Video 3*). In summary, our data demonstrate that the restriction mechanism of PML bodies involves an entrapment of incoming

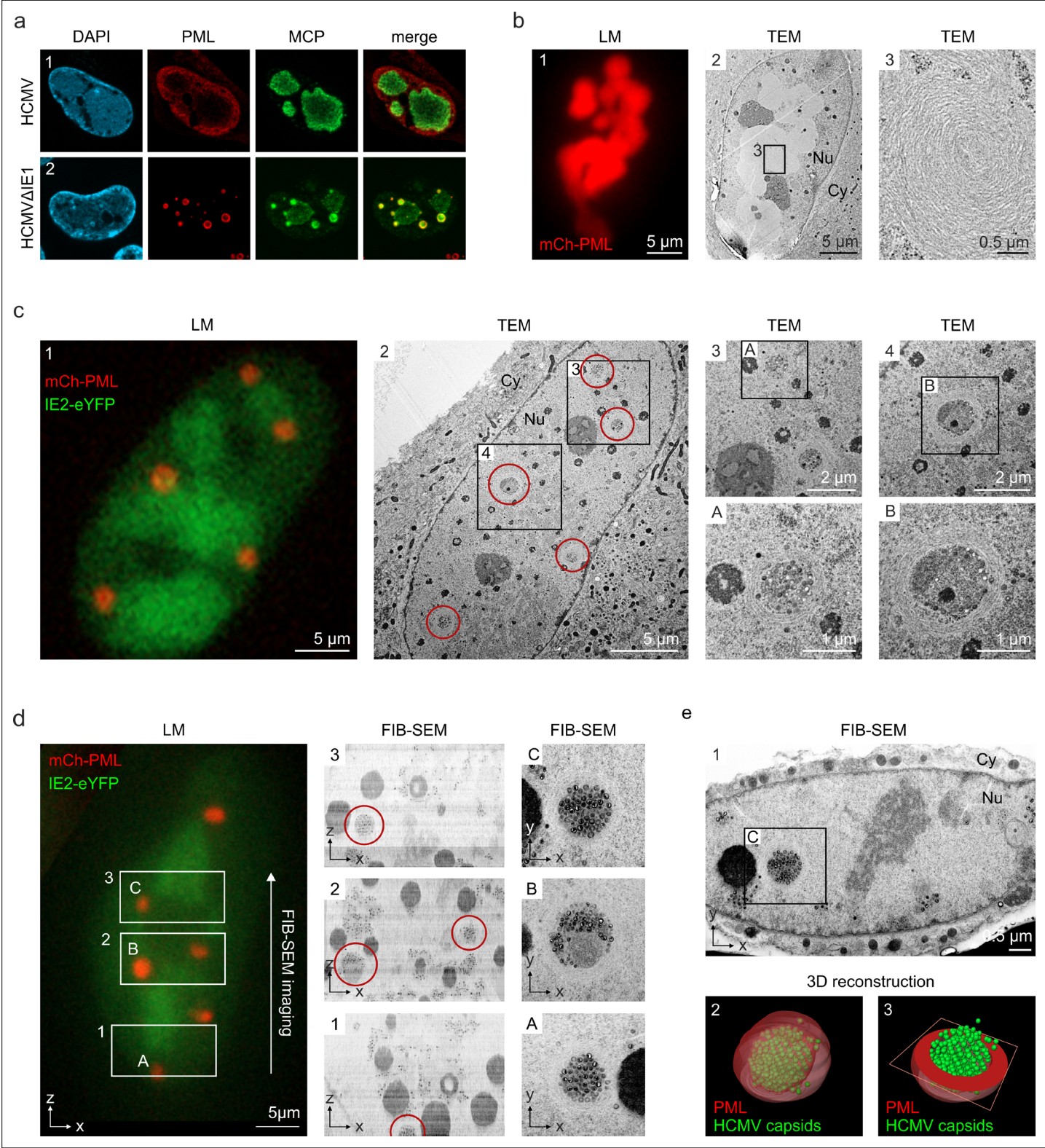

**Figure 7.** Entrapment of HCMV nuclear capsids by PML cages. (**a**) Localization of major capsid protein MCP to PML cages. HFF were infected with HCMV (MOI of 0.5) or HCMVΔIE1 (MOI of 5), based on strain AD169, and harvested at 3 or 6 days after infection, respectively. Endogenous PML and the HCMV protein MCP were detected by immunofluorescence staining; cell nuclei were counterstained with DAPI. (**b–e**) Visualization of HCMV capsid entrapment in PML cages by correlative light and electron microscopy (CLEM). HFF expressing mCherry-PML were seeded on carbon-patterned sapphire discs and, 1 day later, were infected with a TB40E-based recombinant HCMV lacking IE1 and encoding eYFP-tagged IE2 (MOI >30). 6 days

*Figure 7 continued on next page*

*Figure 7 continued*

after infection, cells were imaged by fluorescence microscopy (LM) and subjected to EM sample preparation. Selected non-infected (**b**) or infected (**c, d**) cells were subsequently analyzed by TEM (**b, c**) or FIB-SEM tomography (**d**, see also *Video 2*). Red circles indicate PML cages that were identified in TEM images (**c**, panel 2) or in overview pictures generated from aligned FIB-SEM images (**d**, panels 1–3). (**e**) 3D reconstruction of a HCMV capsid-containing PML cage. Panel 1: FIB-SEM image from the 2nd imaging series containing the PML cage shown in d, panel C. Panel 2: 3D model showing HCMV capsids in green and surrounding PML in red transparent (see also *Video 3*). Panel 3: 3D model showing HCMV capsids in green and a cross-section of the PML envelope in red. TEM, transmission electron microscopy; FIB-SEM, focused ion beam-scanning electron microscopy; Nu, nucleus; Cy, cytoplasm.

The online version of this article includes the following figure supplement(s) for figure 7:

**Figure supplement 1.** Characterization of HFF with doxycycline-inducible expression of mCherry-PML.

**Figure supplement 2.** Even distribution of viral capsids in HCMV-IE2eYFP-infected cell nuclei.

**Figure supplement 3.** Clustering of viral capsids in HCMVΔIE1-infected cell nuclei.

HCMV genomes as well as newly assembled HCMV capsids that, during wild-type virus infection, is antagonized by the regulatory IE1 protein.

## Discussion

Since their discovery, PML-NBs have been topic of intense investigation, both to elucidate the structure and to understand the function of these enigmatic nuclear organelles. The general interest in PML-NBs originated from the tight link between PML-NB integrity and several pathological conditions such as acute promyelocytic leukemia (APL), neurodegenerative diseases as well as viral infections, and resulted in a large number of publications implicating PML-NBs in diverse cellular processes (*Tavalai and Stamminger, 2008*; *Grimwade and Solomon, 1997*; *Yasuda et al., 1999*). These different biochemical activities likely arise from the high number of PML-associated proteins as well as the dynamic nature of PML-NBs, which differ in composition, number, size, and position depending on the cellular state (*Eskiw et al., 2003*; *Weidtkamp-Peters et al., 2008*). Here, we describe unusually

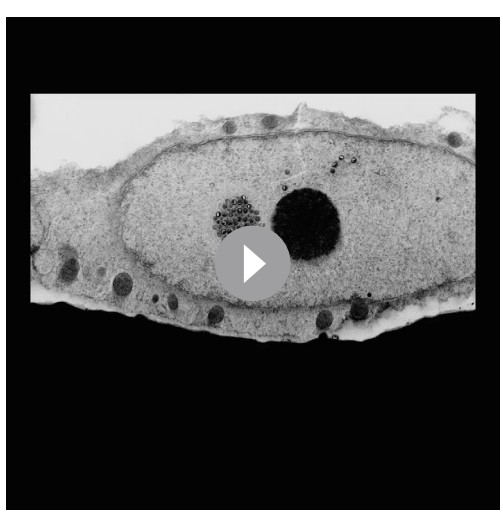

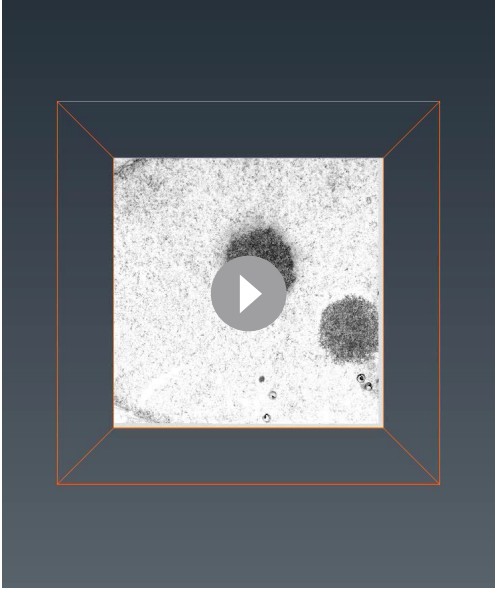

**Video 2.** FIB-SEM tomography showing HCMV capsid entrapment by PML cages. FIB-SEM tomography was applied to analyze three volumes within a mCherry-PML expressing cell that was infected with HCMVΔIE1-IE2eYFP as described in Figure 7d. All regions had the same dimensions (5 μm in z direction) resulting in datasets of 250 subsequent slices with an increment of 20 nm, which were selected and assembled with ImageJ.

https://elifesciences.org/articles/73006/figures#video2

**Video 3.** 3D reconstruction of a PML cage containing HCMV capsids. 3D reconstruction of a PML cage within a cell nucleus infected with HCMVΔIE1-IE2eYFP (see Figure 7d and e, panel c). FIB-SEM images containing the PML cage were assembled and cropped using ImageJ, aligned with IMOD, and segmented in Avizo. HCMV capsids are shown in green, the PML envelope is shown in transparent red.

https://elifesciences.org/articles/73006/figures#video3

large PML-NBs (*Figure 1*), referred to as PML cages, which have the capacity to entrap both parental viral genomes and newly assembled viral capsids thus contributing to the restriction of human cytomegalovirus infection. Concerning biogenesis and composition, PML cages are similar to normal PML-NBs, as the PML protein functions as a key regulator (*Figure 2b*) that recruits all other main components with the only exception of ATRX (*Figure 2a*). Under physiological conditions, ATRX is recruited to PML-NBs via an interaction with the Daxx protein in order to form a chromatin remodeling complex (*Tang et al., 2004*). Upon HCMV infection, Daxx is initially targeted for degradation by the viral tegument protein pp71; however, it re-accumulates at later stages of infection and can be detected at PML cages (*Figure 2a*, *Saffert and Kalejta, 2006*). Since several interaction partners of Daxx, including ATRX, have been shown to bind in a mutually exclusive manner, the absence of ATRX from PML cages may indicate an altered regulation of Daxx during HCMV infection that leads to a switch of interaction partners (*Li et al., 2017*).

In addition to the main PML-NB components, we found heterochromatin protein HP1α localizing to PML cages and being recruited by Sp100 (*Figure 6a and b*), which is in line with a previously described interaction between these two proteins (*Seeler et al., 1998*). The presence of HP1α has also been observed for giant PML bodies in lymphocytes of patients suffering from the ICF syndrome. This PML structure contains a core of cellular satellite DNA, which is present in a hypomethylated and decondensed state in ICF, and therefore suggests a role of PML-NBs in the establishment of condensed heterochomatin (*Luciani et al., 2006*). However, while HP1 proteins localize to the center of the PML body and are surrounded by concentric layers of other PML-NB components, all proteins detected at HCMV-induced PML cages colocalize at the rim thus suggesting subtle differences in the architecture of giant PML-NBs (*Figures 2a and 6a*). Several further reports on the formation of enlarged PML-NBs can be found in literature, for instance as alternative lengthening of telomers (ALT)-associated PML-NBs in telomerase-negative tumor cells, in cells quiescently infected with HSV-1, and during infection with BK virus or Merkel cell polyomavirus (*Everett et al., 2007*; *Jiang et al., 2011*; *Yeager et al., 1999*; *Neumann et al., 2016*). The molecular basis for their formation, however, is far from being understood. Since during HCMV infection, the immediate-early protein IE1 functions as inhibitor of interferon (IFN) signaling, we hypothesized that the drastic enlargement of PML-NBs after infection with IE1-deficient HCMV results from IFN-mediated upregulation of PML-NB proteins (*Paulus et al., 2006*). This would also fit to the observation that PML cages preferentially appear after infection with high doses of IE1-deficient HCMV (*Figure 3a–c*). Indeed, block of the IFN pathway reduces the formation of PML cages suggesting an important role of IFN in the development of these structures and corroborating the previous finding that IFN signaling enhances the antiviral activity of restriction factors (*Figure 4a–c*, *Sakuma et al., 2007*). However, this effect is not just due to increased protein amounts, as overexpression of PML (*Figure 2c*) or stimulation of PML-NB protein expression by IFN treatment (*Figure 4d–f*) is not sufficient to induce PML cages. Our data reveal that induction of a DNA damage response is required as an additional stimulus in HFF cells (*Figure 4g and h*). While most studies that implicate PML-NBs in the DNA damage response have focused on acute cellular DNA damage, very recent data postulate a role in the repair of persistent DNA damage and even show an association of circular PML structures with such DNA lesions (*Imrichova et al., 2019*; *Vancurova et al., 2019*). These PML structures occur after prolonged treatment with DNA damage-inducing agents (*Imrichova et al., 2019*), which fits to our observations (*Figure 4h*) and also to the slow kinetics of PML cages biogenesis in HCMVΔIE1-infected cells (*Figure 1e*). Therefore, it appears likely that PML cages form as a response to continuous interferon and DNA damage signaling during HCMVΔIE1 infection, initiated by incoming viral DNA and enhanced by viral DNA replication, leading to a further enlargement of PML spheres (*Figure 3f*). The requirement of DNA damage signaling for formation of PML cages is supported by our observations that PML cages exhibit a strong association with markers of DNA damage (*Figure 4i, j*) and treatment of HCMVΔIE1 infected cells with an ATM inhibitor significantly reduces the number of giant PML-NBs (*Figure 4k, l*). Of note, IFN signaling was also found to be upregulated in an animal model of ICF syndrome suggesting that dual signaling via the interferon and DNA damage response may constitute a common mechanism whereby entrapment by giant PML nuclear bodies is triggered (*Rajshekar et al., 2018*).

As a major finding, our data provide evidence that PML can enclose both input viral genomes and newly assembled nuclear capsids, which are distinguishable activities occurring at different times of HCMV infection. To study the association of PML cages with incoming HCMV genomes, we made

use of alkyne-modified deoxynucleosides that allow DNA visualization without denaturation and have been successfully utilized to detect adenovirus, vaccinia virus, and herpesvirus DNA (*Wang et al., 2013*). Concerning sensitivity and impact on virus growth, we found that EdC is most suitable for generation of labeled HCMV (*Figure 5—figure supplements 1 and 2a*), although EdU and F-ara-EdU have been reported to be incorporated with higher efficiency in short-pulse experiments or being more sensitive for detection of cellular DNA, respectively (*Manska et al., 2020*; *Neef and Luedtke, 2011*). EdC labeling efficiency of HCMV DNA was determined by costaining of vDNA with tegument protein pp150, serving as a proxy for viral capsids, and was calculated to reach at least 70% (*Figure 5—figure supplement 2c, d*). Visualization of invading HCMV and HCMVΔIE1 genomes at immediate-early times of infection revealed an entrapment of viral DNA (vDNA) in slightly enlarged PML-NBs (*Figure 5a–e*), which is highly efficient in cell nuclei containing only one HCMV genome but significantly reduced with increasing number of viral genomes (*Figure 5a, c*). Thus, our data provide an explanation for the fact that PML-NB-based restriction of HCMV gene expression can only be observed under low MOI conditions and underline the importance of IE1 for initiation of lytic replication at low MOI (*Tavalai et al., 2006*). A similar MOI-dependent envelopment of viral genomes has recently been described for HSV-1 DNA upon nuclear entry (*Alandijany et al., 2018*). Compared to our data on HCMVΔIE1, there appears to be a higher degree of PML-vDNA colocalization indicating a more efficient genome entrapment. However, in contrast to HSV-1, genome entrapment by PML-NBs may already be antagonized prior to de novo viral gene expression by the HCMV tegument protein pp71, which has been shown to disperse ATRX and induce degradation of Daxx. In fact, this scenario is supported by a reduced frequency of PML colocalization with HSV-1 genomes in ATRX-depleted cells (*Alandijany et al., 2018*). Notably, HCMVΔIE1 input genomes remain associated with PML-NBs and can still be detected in a condensed state at the inner rim of PML cages at 3 days after low multiplicity infection (MOI = 0.1) (*Figure 5f*). Although PML cages cannot completely silence HCMV transcription at higher MOIs (MOI = 1), as suggested by genome decondensation and IE2 expression (*Figure 5f*), our data demonstrate a repressive activity because PML disruption under these conditions results in full transcriptional recovery and promotes viral genome replication (*Figure 6c–e*).

As an independent antiviral activity, PML cages are positioned next to viral replication centers and entrap newly assembled viral capsids. By correlative light and electron microscopy (TEM and FIB-SEM tomography), we identify PML cages as circular clusters of viral capsids that are enveloped by PML fibers and occur after high MOI infection with HCMVΔIE1 (*Figure 7c–e*; *Figure 7—figure supplement 3*), but not wild-type HCMV (*Figure 7—figure supplement 2S*). This gives ground to assume that, after initial disruption of PML-NBs, IE1 remains bound to PML in order to antagonize capsid entrapment at late stages of HCMV infection and may explain the metabolic stability of the immediate-early protein (*Scherer et al., 2016*). The question remains of how HCMV genomes and capsids are sensed and entrapped by PML cages. The fact that PML cages form in absence of viral DNA replication implies a formation or incorporation of viral capsids in pre-assembled PML structures rather than an active envelopment of nuclear capsids by PML (*Figure 3e, f*). A similar sequestration of viral capsids by PML cages has been found in cells infected with varicella-zoster virus (VZV) by immuno-electron microscopy (*Reichelt et al., 2011*). However, a unique C-terminal domain in PML isoform IV is required for both VZV capsid binding and antiviral activity, whereas our data show an isoform-independent formation of PML cages (*Figure 2d*). This suggests a requirement of the common N-terminal TRIM region for HCMV restriction and correlates with a previously demonstrated anti-HCMV activity of the shortest, nuclear PML isoform VI (*Tavalai et al., 2006*).

Taken together, we show a multilayered antiviral role of PML-NBs during HCMV infection, which is based on an entrapment mechanism likely leading to a more efficient inhibition or immobilization of viral components. Since the restriction activity of PML at immediate-early times is overcome by high virus doses, entrapment of nuclear capsids may have evolved as an additional line of defense in late infected cells, however, both antiviral strategies are efficiently antagonized by IE1 during wild-type HCMV infection. With regard to previous reports of sequestration events during HSV-1 and VZV infection, we conclude that entrapment of viral components by PML-NBs serves as general mechanism to inhibit herpesviral infections.

# Materials and methods

**Key resources table**

| Reagent type (species) or resource | Designation | Source or reference | Identifiers | Additional information |
|---|---|---|---|---|
| Strain, strain background (*Human cytomegalovirus*) | AD169 | Hobom et al., J. Virol. 2000 PMID:10933677 | | based on BAC HB15 |
| Strain, strain background (*Human cytomegalovirus*) | TB40E | Sinzger et al., J. Gen. Virol. 2008 PMID:18198366 | | based on TB40E-Bac4 |
| Genetic reagent (*Human cytomegalovirus*) | AD169ΔIE1 | Scherer et al., J Virol. 2016 PMID:26559840 | | based on BAC HB15 |
| Genetic reagent (*Human cytomegalovirus*) | AD169/IE1-L174P | Scherer et al., J Virol. 2016 PMID:26559840 | | based on BAC HB15 |
| Genetic reagent (*Human cytomegalovirus*) | TB40E-IE2-EYFP | *Wagenknecht et al., 2015* PMID:26057166 | | based on TB40E-Bac4 |
| Genetic reagent (*Human cytomegalovirus*) | TB40E-ΔIE1-IE2-EYFP | this paper | | Deletion of IE1 from TB40E-IE2-EYFP |
| Cell line (*Homo-sapiens*) | U2OS | ATCC | HTB-96 | |
| Cell line (*Homo sapiens*) | HEK293T | DMSZ | ACC 635 | |
| Antibody | mAb-PML G8 (Mouse monoclonal) | Santa Cruz Biotechnology | Cat# sc-377340 | IF (1:500) |
| Antibody | pAb-PML #4 (Rabbit polyclonal) | Peter Hemmerich, Jena, Germany | | IF (1:500) |
| Antibody | pAb-PML A301-167A (Rabbit polyclonal) | Bethyl Laboratories | Cat# A301-167A RRID: AB_873108 | IF (1:1000), WB (1:5000) |
| Antibody | pAb-PML A301-168A (Rabbit polyclonal) | Bethyl Laboratories | Cat# A301-168A RRID: AB_2284081 | IF (1:1000), WB (1:5000) |
| Antibody | pAb-Sp100 B01 (Mouse polyclonal) | Abnova | Cat# H00006672-B01, RRID:AB_982933 | IF (1:1000) |
| Antibody | pAb-Sp100 GH3 (Rabbit polyclonal) | Hans Will, Hamburg, Germany | | WB (1:10000) |
| Antibody | mAb-Daxx MCA2143 (Mouse monoclonal) | Bio-Rad | Cat# MCA2143, RRID:AB_2088900 | IF (1:400) |
| Antibody | pAb-ATRX H300 (Rabbit polyclonal) | Santa Cruz Biotechnology | sc-15408 (discontinued) | IF (1:500) |
| Antibody | Ab-HP1α 2,616 (Rabbit polyclonal) | Cell Signaling Technology | Cat# 2,616 | IF (1:200) |
| Antibody | pAb-SUMO2/3 ab22654 (Rabbit polyclonal) | Abcam | Cat# ab22654, RRID:AB_2198415 | IF (1:500) |
| Antibody | mAb-SUMO1 (Mouse monoclonal) | Gerrit Praefcke, Langen, Germany | | IF (1:100) |
| Antibody | pAb-phospho-Histone H2A.X 20E3 (Rabbit monoclonal) | Cell Signaling Technology | Cat# 9718, RRID:AB_2118009 | IF (1:500) |
| Antibody | mAB-TRF1 TRF78 (Mouse monoclonal) | Santa Cruz Biotechnology | Cat# sc-56807, RRID:AB_793407 | IF (1:250) |
| Antibody | mAb-IE1 p63-27 (Mouse monoclonal) | Andreoni et al., J. Virol. Methods 1989 PMID:2542350 | | WB (1:100), IF (1:5) |
| Antibody | mAb-UL44 BS 510 (Mouse monoclonal) | Plachter et al., Virus Res. 1992 PMID:1329369 | | WB (1:5000), IF(1:1000) |
| Antibody | mAb-MCP 28–4 (Mouse monoclonal) | Waldo et al., Lancet 1989 PMID:2463443 | | WB (1:2), IF (1:2) |
| Antibody | pAb-IE2 pHM178 (Rabbit polyclonal) | Hofmann et al., J Virol 2000 PMID:10684265 | | WB (1:5000), IF (1:1500) |
| Antibody | pAb-UL84 (Rabbit polyclonal) | Hofmann et al., J Virol 2000 PMID:10684265 | | WB (1:5000), IF (1:1500) |

*Continued on next page*

*Continued*

| Reagent type (species) or resource | Designation | Source or reference | Identifiers | Additional information |
|---|---|---|---|---|
| Antibody | anti-FLAG (Mouse monoclonal) | Sigma-Aldrich | Cat# F1804, RRID:AB_262044 | IF (1:1000) |
| Antibody | anti-Myc (Mouse monoclonal) | produced in hybridoma cells, ATCC | MYC 1-9E10.2 [9E10] | WB (1:10), IF (1:2) |
| Antibody | anti-beta-Actin (Mouse monoclonal) | Sigma-Aldrich | Cat# A5441, RRID:AB_476744 | WB (1:10000) |
| Antibody | anti-phospho-STAT2 (Tyr689) | Millipore | Cat# 07–224 | WB (1: 500) |
| Antibody | anti-Interferon-alpha/beta Receptor Chain 2 (Mouse monoclonal) | Millipore | Cat# MAB1155, RRID:AB_2122758 | (5 µg/ml) |
| Recombinant DNA reagent | pLVX-shRNA1 (plasmid) | Clontech | | lentiviral construct for stable knockdown |
| Recombinant DNA reagent | pLVX-shPML (plasmid) | *Wagenknecht et al., 2015* PMID:26057166 | | lentiviral construct for stable knockdown |
| Recombinant DNA reagent | pLVX-shSp100 (plasmid) | *Wagenknecht et al., 2015* PMID:26057166 | | lentiviral construct for stable knockdown |
| Recombinant DNA reagent | pLVX-shDaxx (plasmid) | *Wagenknecht et al., 2015* PMID:26057166 | | lentiviral construct for stable knockdown |
| Recombinant DNA reagent | pLVX-shATRX (plasmid) | this paper | | cloning of shRNA directed against ATRX into pLVX-shRNA1 |
| Recombinant DNA reagent | pLKO-FLAG-PML I to VI (plasmid) | Cuchet et al., J Cell Sci 2011 PMID:21172801 | | lentiviral constructs for stable expression |
| Recombinant DNA reagent | pInducer20-CRSmut (plasmid) | Schilling et al., PLoS Pathog. 2021 PMID:33770148 | | lentiviral construct for inducible expression, mutated cis-repression signal |
| Recombinant DNA reagent | pInducer20-CRSmut-mCherry-PML VI (plasmid) | this paper | | lentiviral construct for inducible expression, mutated cis-repression signal |
| Recombinant DNA reagent | pLKO-Myc-IE1 (plasmid) | Scherer et al., PLoS Pathog 2014 PMID:25412268 | | lentiviral construct for stable expression |
| Recombinant DNA reagent | pLKO-Myc-IE1 1–382 (plasmid) | Scherer et al., PLoS Pathog 2011 PMID:26559840 | | lentiviral construct for stable expression |
| Chemical compound, drug | Camptothecin; CPT | Santa Cruz Biotechnology | Cat# sc-200871 | |
| Chemical compound, drug | Doxorubicin hydrochloride; Doxo | Cayman Chemical | Cat# Cay15007 | |
| Chemical compound, drug | ATM inhibitor KU 55933; ATMi | Tocris Bioscience | Cat# 3,544 | |
| Chemical compound, drug | 5-Ethynyl-2'-deoxycytidine; EdC | Sigma-Aldrich | Cat# T511307 | |
| Chemical compound, drug | 5-Ethynyl-2'-deoxyuridine; EdU | Sigma-Aldrich | Cat# 900,584 | |
| Chemical compound, drug | (2'S)–2'-Deoxy-2'-fluoro-5-ethynyluridine; F-ara-EdU | Sigma-Aldrich | Cat# T511293 | |

## Oligonucleotides and plasmids

The oligonucleotide primers used for this study were purchased from Biomers GmbH (Ulm, Germany) and are listed in *Supplementary file 1*. Lentiviral pLVX-shRNA1-based vectors containing control shRNA or shRNAs directed against PML, Sp100, Daxx, and ATRX (pLVX-shRNA1, pLVX-shPML/shSp100/shDaxx/shATRX) were generated as described previously (see *Supplementary file 1* for target sequences) (*Wagenknecht et al., 2015*). For stable overexpression of FLAG-tagged PML isoforms, lentiviral pLKO-based vectors were used (pLKO-FLAG-PML I to VI), which were kindly provided by Roger Everett (Glasgow, United Kingdom) (*Cuchet et al., 2011*). The lentiviral vector used for doxycycline-inducible expression of mCherry-PML isoform VI (pInducer20-CRS-mut-mCherry-PML VI), was generated via PCR amplification of mCherry-PML from pHM2396 using primer 5'attB1-mCherry and 3_attB3-PMLVI (*Tavalai et al., 2006*). The PCR product was inserted into pInducer20-CRSmut via

a combined BP/LR Gateway recombination reaction utilizing pDONR221 as intermediate vector (Invitrogen, Thermo Fisher Scientific Inc, Waltham, MA, USA). The pInducer20-CRSmut vector was established by site-directed mutagenesis of the cis-repression sequence (CRS) within the promoter region of pInducer20 (a gift from S. Elledge) with primers c-CRS-mut and nc-CRS-mut (see *Supplementary file 1*), since the CRS leads to transcriptional repression during HCMV infection (*Meerbrey et al., 2011*; *Reuter et al., 2018*). Lentiviral pLKO-based vectors encoding Myc-tagged IE1 and IE1 1–382 (pLKO-Myc-IE1, pLKO-Myc-IE1 1–382) were described elsewhere (*Scherer et al., 2014*).

## Cells and viruses

Primary human foreskin fibroblast (HFF) cells were isolated from human foreskin tissue and cultivated at 37 °C and 5% $CO_2$ in Eagle's minimal essential medium (Gibco, Thermo Fisher Scientific Inc, Waltham, MA, USA) containing 7% fetal calf serum (FCS) (Sigma-Aldrich, Merck KGaA, Darmstadt, Germany), 1% GlutaMAX (Gibco), and penicillin-streptomycin (Sigma-Aldrich). HFF cells with a stable knockdown of PML, Sp100, hDaxx or ATRX were maintained in HFF cell culture medium additionally supplemented with 5 µg/ml puromycin (InvivoGen, San Diego, CA, USA). HFF expressing FLAG-tagged PML isoforms or mCherry-tagged PML were cultured in the presence of 500 µg/ml geneticin (InvivoGen) and 10% FCS, which induced sufficient expression of mCherry-PML (*Figure 7—figure supplement 1*). HEK293T (DMSZ ACC 635) and U2OS (ATCC HTB-96) cells were cultivated in Dulbecco's minimal essential medium (DMEM) containing Glutamin (Gibco) and supplemented with 10% FCS (Sigma-Aldrich) and penicillin-streptomycin (Sigma-Aldrich). Identity of cell lines has been confirmed by short tandem repeat typing. Cells were tested for absence of mycoplasma contamination using a commercially available mycoplasma detection kit (MycoAlert, Lonza).

Infection experiments of HFF cells were performed with the HCMV strain AD169 as well as recombinant viruses AD169ΔIE1, AD169/IE1-L174P, TB40E-IE2eYFP, and TB40E-ΔIE1-IE2eYFP at defined multiplicities of infection (MOI). To determine titers of AD169 and TB40E-IE2eYFP, HFF were infected with serial dilutions of virus supernatants and, after 24 hr of incubation, cells were fixed and stained for IE1. Subsequently, the number of IE1-positive cells was determined, and viral titers were calculated (IE units/ml). IE1-deleted/mutated viruses were titrated via determination of genome equivalents in infected cells. For this, HFF cells were infected with various dilutions of wild-type and recombinant viruses. Sixteen hr later, viral and cellular DNA was extracted from infected cells using the DNeasy Blood & Tissue Kit (Qiagen, Hilden, Germany) and subjected to quantitative real-time PCR of HCMV gB and cellular albumin as described below. Since infection of HFF cells with equivalent genome copy numbers of wild-type and IE1-deficient HCMV resulted in comparable levels of IE protein IE2p86, MOIs are indicated in IE units per cell (IEU/cell) for all viruses (*Scherer et al., 2016*).

For infection experiments, HFF were seeded into six-well dishes at a density of $3 \times 10^5$ cells/well. One day later, cells were incubated with 1 ml of virus suspension for 1.5 hr and were provided with fresh medium, before they were used for subsequent western blot or immunofluorescence analyses. For correlative light and electron microscopy (CLEM), HFF were seeded at a density of $4 \times 10^4$ cells/well on carbon coated and glow discharged sapphire discs (Wohlwend GmbH) with coordinate system that were placed into the wells of an 8-well µ-slide (ibidi GmbH, Gräfelfing, Germany). After 24 hr, cells were incubated with 300 µl of viral supernatant for 1.5 hr, before the virus suspension was replaced with fresh medium. Infectivity of TB40ΔIE1-IE2eYFP was enhanced by centrifugation at 2000 rpm for 10 min at room temperature.

## Generation of recombinant cytomegaloviruses

Recombinant cytomegaloviruses AD169ΔIE1 and AD169/IE1-L174P are based on bacterial artificial chromosome (BAC) HB15 and were published previously (*Scherer et al., 2016*). The recombinant viruses TB40E-IE2eYFP and TB40EΔIE1-IE2eYFP were generated by recombination-based genetic engineering of HCMV BAC TB40E-Bac4 (kindly provided by Christian Sinzger, Ulm, Germany). For fusion of eYFP to the coding region of IE2, resulting in TB40E-IE2eYFP, markerless 'en passant' mutagenesis was performed as described (*Wagenknecht et al., 2015*; *Tischer et al., 2006*). Briefly, a linear recombination fragment was generated by amplification of an I-SceI-aphAI cassette from plasmid pHM3366 (universal transfer construct based on pEPkan-S) using primers IE2-eYFP-forw and IE2-eYFP-rev (*Supplementary file 1*). For homologous recombination, the PCR fragment was treated with *DpnI*, gel purified, and transformed into *Escherichia coli* strain GS1783 harboring TB40E-Bac4 (a gift of

M. Mach, Erlangen), before two-step bacteriophage $\lambda$ red-mediated recombination was conducted (*Tischer et al., 2006*). Positive transformants were identified by growing the bacteria on agar plates containing kanamycin (first recombination) or chloramphenicol and 1% arabinose (second recombination) at 32 °C for 1 or 2 days. In order to obtain TB40ΔIE1-IE2eYFP that, in addition to the eYFP-IE2 fusion protein, harbors a deletion of IE1 exon 4, we utilized the Red-mediated recombination method published by *Datsenko and Wanner, 2000*. For this purpose, a linear recombination fragment, which comprises a kanamycin resistance cassette as well as 5′ and 3′ genomic sequences, was produced by PCR amplification from pKD13 using primers 5′Intron3/pKD13 and 3′Exon4/pkd13 (*Supplementary file 1*). This fragment was used for transformation of electrocompetent *Escherichia coli* strain DH10B harboring the TB40E BAC, and homologous recombination was performed as described previously (*Datsenko and Wanner, 2000*). After every recombination step, BAC DNA was purified from bacterial colonies and successful recombination was confirmed by restriction fragment length polymorphism analysis (RFLP), PCR, and sequencing.

HFF and HFF stably expressing IE1 (*Scherer et al., 2014*) were utilized for reconstitution of TB40E-IE2eYFP and TB40ΔIE1-IE2eYFP, respectively. The cells were seeded in six-well dishes at a density of $3 \times 10^5$ cells/well, followed by co-transfection with 1 µg purified BAC DNA, 0.5 µg pp71 expression plasmid pCB6-pp71 and 0.5 µg of a vector encoding the Cre recombinase using FuGENE6 transfection reagent (Promega, Madison, WI, USA). Transfected HFF were cultivated until viral plaque formation was observed and the supernatants from these cultures were used for further virus propagation in HFF or IE1-expressing HFF, before the cell culture supernatant was centrifuged to remove cellular debris and stored at –80 °C in aliquots.

## Stable knockdown and overexpression of proteins in HFF by lentiviral transduction

Lentiviral transduction was used to establish HFF with a stable knockdown of PML-NB proteins, for HFF that stably overexpress FLAG-tagged PML isoforms, HFF with inducible expression of mCherry-PML, and for expression of IE1. Replication-deficient lentiviruses were produced in HEK293T cells, which were seeded in 10 cm dishes at a density of $5 \times 10^6$ cells/dish. After 24 hr, cells were transfected with the respective lentiviral vector together with packaging plasmids pLP1, pLP2, and pLP/VSV-G using the Lipofectamine 2000 reagent (Invitrogen). Sixteen hr later, cells were provided with fresh medium and 48 hr after transfection, viral supernatants were harvested, filtered through a 0.45 µm sterile filter, and stored at –80 °C. To transduce primary HFF, the cells were incubated for 24 hr with lentivirus supernatant in the presence of 7.5 µg/ml polybrene (Sigma-Aldrich). Stably transduced cell populations were selected by adding 5 µg/ml puromycin or 500 µg/ml geneticin to the cell culture medium.

## Quantitative real-time PCR

To quantify intracellular viral genome copies, total DNA was extracted from infected cells using the DNeasy Blood & Tissue Kit (Qiagen, Hilden, Germany) and subjected to quantitative qPCR amplification of an HCMV gB (UL55)-specific target sequence and a sequence region in the cellular albumin gene (ALB) as reference gene (see *Supplementary file 1* for sequences of primers and hydrolysis probes). For determination of reference $C_T$ values (cycle threshold), serial dilutions of the respective standards ($10^8$–$10^2$ DNA molecules of HCMV UL55 or cellular ALB) were examined by PCR reactions in parallel. Real-time PCR was conducted using an Applied Biosystems 7,500 Real-Time PCR System (Applied Biosystems, Foster City, CA, USA) as described previously or an Agilent AriaMx Real-time PCR System with the corresponding software AriaMx 1.5 (Agilent Technologies, Inc, Santa Clara, CA, USA) (*Scherer et al., 2016*). The 20 µL reaction mix contained 5 µL sample or standard DNA together with 10 µL 2 x SsoAdvanced Universal Probes Supermix (Biorad), 1 µL of each primer (5 µM stock solution), 0.3 µL of probe (10 µM stock solution), and 2.7 µL of $H_2O$. The thermal cycling conditions consisted of an initial step of 3 min at 95 °C followed by 40 amplification cycles (10 s at 95 °C, 30 s 60 °C). Finally, genome copy numbers were calculated with the sample-specific $C_q$ value set in relation to the standard serial dilutions.

## Western blotting

For western blot analysis, total cell lysates were prepared in a sodium dodecyl sulfate-polyacrylamide gel electrophoresis (SDS-PAGE) loading buffer by boiling for 10 min at 95 °C followed by sonication

for 1 min. Proteins were separated on SDS-containing 8–12% polyacrylamide gels and transferred to PVDF membranes (Bio-Rad Laboratories, Inc, Hercules, USA). After staining with primary and secondary antibodies, proteins were visualized by chemiluminescence detection using a LAS-1000plus image analyzer (Fuji Pharma, Tokyo, Japan) or a FUSION FX7 imaging system (Vilber Lourmat, Eberhardzell, Germany).

## Immunofluorescence analysis

Indirect immunofluorescence analysis of HFF cells was performed by fixation with 4% paraformaldehyde and fluorescence staining as described elsewhere (*Scherer et al., 2014*). When late stages of HCMV infection were investigated, an additional blocking step was performed by incubating cells with 2 mg/ml γ-globulins from human blood (Sigma) for 30 min at 37 °C, before primary antibodies were applied. Confocal images were obtained with a Leica TCS SP5 confocal microscope or a Zeiss Axio Observer Z1 equipped with an Apotome.2. The images were processed with Adobe Photoshop CS5 or ZEN 2.3 and assembled using CorelDraw. For 3D reconstruction of z-series images, Huygens Professional Software (Scientific Volume Imaging) was used. Stacks of confocal images from single cells were imaged, deconvoluted using the Huygens Deconvolution wizard and 3D images generated with the Huygens Surface Renderer tool. For quantifications, z-series images (0.3–0.7 µm distance) of at least 50 cell nuclei per sample were taken. To measure number and size (perimeter) of PML foci, automated ImageJ-based quantification was performed on maximum intensity projection images 3D animations were generated from z-series images using the Leica LAS AF software.

## Antibodies and chemicals

Following antibodies were used to detect PML and associated proteins: mAb-PML G8 (Santa Cruz Biotechnology, Inc, Dallas, TX, USA), pAb-PML #4 (provided by P. Hemmerich, Jena, Germany), pAb-PML A301-167A in combination with pAb-PML A301-168A (Bethyl Laboratories, Montgomery, TX, USA), pAb-Sp100 B01 for IF (Abnova, Taipeh, Taiwan), pAb-Sp100 GH3 for WB (provided by H. Will, Hamburg, Germany), mAb-Daxx MCA2143 (Bio-Rad), pAb-ATRX H300 (Santa Cruz), Ab-HP1α 2,616 (Cell Signaling Technology, Inc Danvers, MA, USA), pAb-SUMO2/3 ab22654 (Abcam, Cambridge, UK), mAb-SUMO1 that was provided by Gerrit Praefcke (Langen, Germany), and pAb-phospho-Histone H2A.X 20E3 (Cell Signaling Technology). TRF1 was detected with mAB-TRF1 TRF78 obtained from Santa Cruz. Viral proteins were detected with mAb-IE1 p63-27 (*Andreoni et al., 1989*), mAb-UL44 BS 510 (provided by B. Plachter, Mainz, Germany), and mAb-MCP 28–4 (*Waldo et al., 1989*). Polyclonal antisera against viral proteins IE2 (pAb-IE2 pHM178) and UL84 (pAb-UL84) were described previously (*Hofmann et al., 2000*). FLAG and Myc tagged-proteins were detected with mAb-FLAG M2 (Sigma-Aldrich) and mAb-Myc 1-9E10.2 (ATCC), respectively. β-actin was detected with MAb β-actin AC-15 from Sigma-Aldrich. Horseradish peroxidase-conjugated anti-mouse/-rabbit secondary antibodies for Western blot analysis were obtained from DIANOVA GmbH (Hamburg, Germany). Alexa Fluor 488-/555-/647-conjugated secondary antibodies for indirect immunofluorescence experiments were purchased from Invitrogen. PAb directed against phospho-STAT2 (Tyr689) and MAb directed against human IFNα/β receptor chain 2 (MAB1155) were purchased from Merck-Millipore.

Camptothecin (sc-200871) and doxorubicin hydrochloride (Cay15007-5) were obtained from Santa-Cruz and Biomol (Biomol GmbH, Hamburg, Germany), respectively and dissolved in DMSO. ATM inhibitor Ku 55,933 (3544) was purchased from Tocris Bioscience (Bristol, UK) and dissolved in DMSO. The ethynyl-labeled nucleosides 5-Ethynyl-2'-deoxyuridine (EdU), 5-Ethynyl-2'-deoxycytidine (EdC), and (2'S) –2'-Deoxy-2'-fluoro-5-ethynyluridine (F-ara-EdU) were purchased from Sigma-Aldrich, dissolved in water, and used at indicated concentrations.

## Labeling of viral DNA with ethynyl-modified nucleosides and detection by click chemistry

In order to visualize viral DNA synthesis during infection, HFF cells were treated with EdU/EdC/F-ara-Edu at 48 hr post-infection and fixed 24 hr later for staining with fluorescent azides and antibodies as described below. In order to produce labeled virus stocks, HFF or HFF-IE1 cells were infected with AD169 or AD169ΔIE1 (MOI of 1), respectively, and EdU/EdC was added at a final concentration of 0.1 µM or 1 µM at 24 hr post-infection. Infected cell cultures were treated with fresh EdU/EdC every 24 hr until a strong cytopathic effect was observed. Supernatants from infected cells were clarified

by centrifugation at 3000 rpm for 10 min and then pelleted by ultra-centrifugation at 23,000 rpm for 70 min at 10 °C. Pellets were carefully rinsed with MEM, before they were resuspended using a 20 gauge syringe needle and filtered through a 0.45 µm sterile filter.

To determine the efficiency of EdU/EdC labeling, HFF cells seeded on coverslips were incubated with viral supernatants for 20 min, washed twice, and supplemented with fresh medium. At 6 hpi, cells were fixed with acetone for 5 min and stained for the HCMV tegument protein pp150 as marker for viral particles. After washing cells twice with PBS, a copper-catalyzed click reaction was performed for detection of viral DNA by incubating the cells for 90 min at RT with freshly prepared labeling solution containing 10 µM Alexa Fluor 488 Azide (Invitrogen), 10 mM sodium ascorbate, 1 mM copper (II) sulfate, 10 mM aminoguanidine, and 1 mM THPTA in PBS. Cells were washed twice with PBS for 2 min and once for 30 min, before coverslips were mounted on microscope slides using Vectashield Antifade Mounting Medium with DAPI (Vector laboratories, Maravai LifeSciences, San Diego, CA, USA) and sealed with nail polish. Pp150 signals that were located at the cell nuclei as well as viral DNA signals in the same nuclei were counted in order to calculate the EdU/EdC labeling efficiency.

In order to visualize viral DNA in combination with PML or γH2AX antibody staining, HFF infected with EdC/EdU-labeled virus were fixed with 4% PFA for 10 min and quenched with 50 mM ammonium chloride and 50 mM glycine in PBS for 5 min at RT. Afterwards, cells were washed twice with PBS, permeabilized with 0.2% TritonX100 in PBS for 15 min at 4 °C, and stained with antibodies as described (*Scherer et al., 2014*). After washing cells twice with PBS, detection of the viral DNA by click chemistry was performed as described above.

## Correlative light and electron microcopy (CLEM)

For CLEM analysis, sapphire disks with infected cells were prepared as described above (see Cells and viruses) and imaged by live-cell fluorescence microscopy before being subjected to EM sample preparation. For fluorescence imaging, the cell culture medium was replaced with Leibovitz's L-15 medium and whole sapphire disks were imaged at 37 °C with a 20 x lens objective of a Zeiss Axio Observer Z1 using the tiling and stitching functions. For EM sample preparation, infected HFFs grown on sapphire discs were fixed by high-pressure freezing (HPF Compact 01; Wohlwend GmbH) followed by freeze substitution and stepwise embedding in epoxy resin (Sigma-Aldrich) as described (*Walther and Ziegler, 2002*; *Villinger et al., 2012*). The embedded cells were first visualized in an inverted light microscope, compared with fluorescence images and areas for TEM and FIB-SEM analysis were selected. For TEM analysis, ultrathin sections of 70 nm thickness were prepared, mounted on formvar coated single slot grids (Plano, Wetzlar, Germany) and examined with a JEM-1400 (Jeol) TEM at an accelerating voltage of 120 kV. TEM images were processed with Photoshop Elements 2018 and assembled with CorelDraw 2018. FIB-SEM analysis was conducted as described previously (*Villinger et al., 2015*). In short, a resin disc containing the embedded cells (height of ~1 mm) was mounted onto a SEM specimen stub. The sample was coated with 5 nm of platinum using an electron beam evaporator (Baltec, Balzers, Liechtenstein) and FIB-SEM tomography was conducted with a Helios Nanolab 600 (FEI, Eindhoven, The Netherlands). The coordinate system was used for localization of the selected cells. Contours of the embedded cells were visualized at an acceleration voltage of 10 kV. In order to protect the selected cells from beam damage during FIB-milling, the area was covered with an additional platinum layer using ion beam-induced platinum deposition. A block face was then generated to gain access to the internal structures of the selected cell. Slice and view was performed using the software module Auto Slice & View.G1 (FEI). With each step, 20 nm of material was removed by the FIB and the newly produced block face was imaged with the SEM at an accelerating voltage of 5 kV using the secondary electron signal recorded with the through-the-lens detector for TEM-like resolution (*Villinger et al., 2012*). The nominal increment of 20 nm between two images was chosen considerably smaller than the diameter of a capsid so that every capsid could be detected.

The open source software IMOD (*Kremer et al., 1996*) was used for automatic alignment of FIB-SEM images. Capsids and PML cages were segmented manually in Avizo 9.4.0 (Thermo Fisher Scientific). Videos were generated with Avizo 9.4.0 or ImageJ and processed with VSDC Video Editor.

## Statistical analysis

Statistical analyses were performed with the build-in data analysis functions of Microsoft Excel. Datasets with more than two groups of data were first analyzed by one-way-ANOVA to test for significant

differences between the various conditions. If the ANOVA indicated significant differences within the dataset, post-hoc analyses were conducted to compare each condition individually with the control group by using unpaired t-tests. p-values < 0.05 were considered marginally significant, < 0.01 significant, and <0.001 highly significant.

## Acknowledgements

This work was supported by the Deutsche Forschungsgemeinschaft (grant STA357/7-1 and grant STA357/8-1). 3D reconstruction of confocal images was performed with the support of Dr. Benjamin Schmid of the Optical Imaging Centre Erlangen (OICE). We would like to thank Andrea Bauer (Ulm) for help with 3D reconstruction in Avizo.

## Additional information

### Funding

| Funder | Grant reference number | Author |
|---|---|---|
| Deutsche Forschungsgemeinschaft | STA357/7-1 | Thomas Stamminger |
| Deutsche Forschungsgemeinschaft | STA357/8-1 | Thomas Stamminger |

The funders had no role in study design, data collection and interpretation, or the decision to submit the work for publication.

### Author contributions

Myriam Scherer, Conceptualization, Data curation, Formal analysis, Investigation, Methodology, Validation, Visualization, Writing – original draft, Writing – review and editing; Clarissa Read, Data curation, Formal analysis, Investigation, Methodology, Validation, Visualization, Writing – original draft; Gregor Neusser, Formal analysis, Investigation, Methodology, Software; Christine Kranz, Regina Müller, Florian Full, Sonja Wörz, Paul Walther, Investigation, Methodology; Anna K Kuderna, Formal analysis, Investigation, Methodology; Anna Reichel, Formal analysis, Investigation, Methodology, Validation, Visualization; Eva-Maria Schilling, Data curation, Investigation, Methodology, Validation; Thomas Stamminger, Conceptualization, Funding acquisition, Investigation, Project administration, Resources, Supervision, Writing – original draft, Writing – review and editing

### Author ORCIDs

Thomas Stamminger http://orcid.org/0000-0001-9878-3119

### Decision letter and Author response

Decision letter https://doi.org/10.7554/eLife.73006.sa1
Author response https://doi.org/10.7554/eLife.73006.sa2

## Additional files

### Supplementary files

• Supplementary file 1. Sequences of oligonucleotides used in this study.

• Transparent reporting form

### Data availability

All data generated or analyzed during this study are included in the manuscript and supporting file; Source Data files have been provided for Figures 1, 3, 4, 5, 6 and Figure 5-figure supplement 2.

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
