## [Editor Report]

This work adds to our understanding of cellular structures called promyelocytic leukaemia nuclear bodies (PML-NBs) in different contexts. The authors show that PML, the principal scaffolding protein of PML-NBs, forms a variety of different structures in response to viral infection, immune stimulation, and DNA damage. Further, they identify PML to restrict the replication of human cytomegalovirus (HCMV) at multiple stages of infection through the formation of alternate PML-scaffold assemblies, which gives important new insights into the interaction between this opportunistic viral pathogen and its host.

---

## [Decision Letter]

**Decision letter after peer review:**

Thank you for submitting your article "Dual signaling via interferon and DNA damage response elicits entrapment by giant PML nuclear bodies" for consideration by *eLife*. Your article has been reviewed by 3 peer reviewers, and the evaluation has been overseen by a Reviewing Editor and Päivi Ojala as the Senior Editor. The following individual involved in review of your submission has agreed to reveal their identity: Marina Lusic (Reviewer #1).

Essential revisions:

All reviewers agreed that your work is highly interesting, but also agreed on the necessity to address the question of DNA damage involvement in the formation of PML cages in the context of HCMV infection. In addition, they all noted that quantifications are necessary for proper interpretation of the data, specifically quantitation of input genomes and statistical population analysis.

*Reviewer #1 (Recommendations for the authors):*

The formation of giant PML bodies in response to IE1 deficient HCMV infection is well supported by the compelling microscopy data showing the entrapment of viral genomes within these structures. The aspects related to the importance of viral titer are very interesting, as it seems that these structures are efficiently entrapping viral genomes (as well as viral capsids at different time points) only at the lower MOIs.

However, judging from the title, the authors might have aimed at showing that IFN signaling and DNA damage play an important role in formation of these cages, as they can be created in cells even in the absence of viral infection. Although their data do suggest that IFN signaling together with DNA damage response plays a perhaps more important role in the initiation of PML accumulation, these observations were not further exploited and some additional controls could allow for a better understanding of the underlying mechanism.

The authors treat HFF with campthotecin or doxyrubicin (inhibitor of TopoI), but do not show any proof of DNA damage. A staining with γH2AX (or TP53BP1) would be helpful to understand if these bodies are formed in the proximity of DNA damage sites.

Another point that I found would be very interesting to address is the turnover of PML within the CAGE like structures. Are these giant PML bodies susceptible to arsenic trioxide degradation? Arsenic has a well documented role in PML turnover and degradation, and is also used as therapeutic agent in treatment of APL. Moreover, it has also been used in clinical trials for the treatment of HTLV, as well as of HIV-1. It would therefore be very interesting to see if the PML can be degraded by arsenic withing these CAGE like structures, and if this has a relevance for viral reactivation.

*Reviewer #2 (Recommendations for the authors):*

This reviewer recognises the strength of this submission and its contribution to the field. The points below aim to strengthen the study further to provide greater clarity and context to a non-specialist reader.

1. The authors should demonstrate the percentage of viral genomes labelled in their HCMV.EdC virion stock. Based on analogous data derived from HSV-1 (Alandijani et al., 2018), EdC genome labelling efficiency within any given stock of particles can vary. Thus, there may be a significant population of unlabelled HCMV genomes within infected cells that are undetectable by click chemistry. This population should be quantified and presented as supplemental data, as this population of unlabelled genomes may significantly influence the impact and interpretation of data and statements presented in the manuscript. For example lines 271 to 276, 'However, by far not all PML cages contained HCMV genomes suggesting that their enlargement is not a direct consequence of the entrapment process. Taken together, out data provide evidence that genome entrapment by PML-NBs has evolved as a mechanism to achieve efficient and persistent repression of HCMV. Reorganization from dot-like foci to PML cages, however, cannot be directly linked to vDNA entrapment but seems to result from a more general stimulus'.

This statement holds true if 90-100% of genomes delivered to the nucleus are detectable by click chemistry. However, if the percentage of genomes labelled is {less than or equal to} 90%, then the authors cannot make this conclusion, as vDNA undetectable by click chemistry may contribute to the phenotypes they observe. Similarly, the authors should make attempts to quantify the precise genome load delivered to cells at the indicated multiplicities of infection used (currently derived from FFU titration). HCMV virus preparations are well established to have a high genome to PFU ratios. Accurate quantitation will provide greater context to non-specialists as to what low or high MOI means by providing absolute numbers for the indicated MOI.

2. One of the driving conclusions presented by the authors is the requirement for the DDR pathway to be activated to form PML-cages. While the authors present compelling data for this in the context of drug induced DNA damage on uninfected cells (Figure 4g), the authors fail to make the same connection during HCMV infection. The authors should examine the localization of PML-cages induced during deltaIE1-HCMV infection to that of γ-H2Ax, a marker of DNA damage. If their hypothesis holds true, PML-cages should localize in proximity to sites of cellular chromatin damage, thereby providing supporting evidence to one of the papers major conclusions. Alternatively, the authors should investigate the influence of DDR inhibitors (e.g. ATM, ATR, and DNA-PKcs inhibitors) on the frequency PML-cage formation to provide the supporting evidence required to make this conclusion stand during HCMV infection. Correspondingly, if vDNA replication is not required for PML-cage assembly, when is the DDR pathway activated during HCMV infection to promote PML-cage assembly?

3. The manuscript lacks detailed quantitation in its image analysis. While some experiments are quantified, many of the figures (e.g. figures 1 and 2) only show representative images. This makes it hard for the reader to determine the relative population of cells that induce the assembly of PML-cages within any given population of infected cells. Is this phenotype rare or frequent within infected cells? It is recommended that all immunostaining experiments are quantified for readership clarity over a minimum of three independent biological replicates (see below).

*Reviewer #3 (Recommendations for the authors):*

A. Major items to be addressed

General:

– The reviewer would suggest, to strengthen the physiological relevance of the formation of the PML cages around HCMV genomes and capsids, to perform, if possible, some of the experiments in bone marrow CD34+ cells and/or THP1 cells (already used by the team as HCMV model of latency in previous studies). This, to reviewer's view, could improve significantly, and beyond the molecular aspects, the understanding of the role of the PML NBs in the control of HCMV infection and their role in the establishment of latency and abortive productive infection through the formation of PML cages.

– l446-448: In the discussion the authors state: "The fact that PML cages form in absence of viral DNA replication implies a formation or incorporation of viral capsids in pre-assembled PML structures rather than an active envelopment of nuclear capsids by PML (Figure 3e, f). » This sentence mingles two aspects of the study: (1) The formation of PML cages around incoming viral genomes; and (2) the formation of PML cages containing viral capsids. To reviewer's opinion the statement, as written, does not consider the phase separation properties of the PML protein (Banani et al., 2016), and for reviews see (Lallemand-Breitenbach and Thé, 2018; Corpet et al., 2020), and the fact that 90% of PML protein is not associated with PML NBs, and has a diffuse nuclear localization (see Lallemand-Breitenbach and Thé, 2010) and/or participating to chromatin dynamics (for a review see Delbarre and Janicki, 2021). The reviewer would suggest to clarify this statement, which is, to reviewer's opinion, misleading on the real capacity of PML NBs to form de novo around genome loci (see Everett and Murray. ND10 Components Relocate to Sites Associated with Herpes Simplex Virus Type 1 Nucleoprotein Complexes during Virus Infection, JVI, 2005, 10.1128/jvi.79.8.5078-5089.2005), and/or viral capsids. May be the authors should consider the most likely possibility that it could exist different mechanisms for the formation of PML cages entrapping incoming viral genomes, and PML cages entrapping capsids.

References

Banani SF, Rice AM, Peeples WB, Lin Y, Jain S, Parker R and Rosen MK (2016) Compositional Control of Phase-Separated Cellular Bodies. Cell 166: 651-663

Corpet A, Kleijwegt C, Roubille S, Juillard F, Jacquet K, Texier P and LOMONTE P (2020) PML nuclear bodies and chromatin dynamics: catch me if you can! Nucleic Acids Res 389: 251

Delbarre E and Janicki SM (2021) Modulation of H3.3 chromatin assembly by PML: A way to regulate epigenetic inheritance. Bioessays: 2100038

Lallemand-Breitenbach V and Thé H de (2010) PML nuclear bodies. Cold Spring Harbor perspectives in biology 2: a000661

Lallemand-Breitenbach V and Thé H de (2018) PML nuclear bodies: from architecture to function. Curr Opin Cell Biol 52: 154-161

Specific

– The reviewer acknowledges the importance of IFN signaling and DNA damage for the formation of PML cages, caps or forks in non-infected cells. The requirement of IFN signaling for the formation of PML cages in HCMV-infected cells is clearly shown. However, it is not clear in reviewer's mind how DNA damage provoked by the injection of viral genomes in the nucleus is required to lead to the formation of PML cages specifically around the viral genomes. Does the authors infer that viral genomes themselves would sustain damage that would lead to aggregation of PML around those viral genomes? To reviewer's view the requirement of DNA damage for the formation of PML cages specifically around HCMV genome is not definitively proven. The reviewer would suggest to the authors to perform a γ-H2A.X and PML staining together with click chemistry for the visualization of the genomes at different times post infection to see if PML cages-entrapped viral genomes could at a particular stage of the entry in the nucleus be associated with DNA damage. This could then strengthen the discussion about that aspect.

B. Other

Specific

– What is the fate of the capsids that are entrapped in the PML cages? Are they degraded? Do their remain as an abortive, latency-prone, productive infection? Although the reviewer acknowledges the fact that performing such experiments is beyond the scoop of this study, the reviewer would be grateful to the authors to have their feedback on that question. May be this thought could be part of the discussion.

– During infection of cells supporting HCMV latency what is the status of IE1 expression? In other words, does it exist a physiological situation of infection by HCMV where IE1 cannot be produced, and consequently cannot induce the dispersal of PML NBs? This should be discussed.

– Figure 4g: Could the caps and/or forks in non-infected cells be associated to nucleoli or specific chromosome loci?

- Figure 6c,d,e: The impact of IE1 expression on the abrogation of HCMV repression leads to the increase of viral genomes and to the expression of at least UL44. IF data to visualize UL44 as a marker of the viral replication compartments and/or MCP for capsids in IE1 expressing cells should be provided in addition to the quantification data.

– Figure 6 f and g – The low resolution of the pdf combined to the size of the images make it difficult to conclude anything on the disruption of the APB by IE1. TRF1 is hardly visible in figure 6f. In addition the reviewer would recommend to provide a TRF1 and IE1 staining in figure 6g in addition to the PML/IE1. Indeed, APB are formed by the grouping of several telomeres in the APBs visualized by IF by larger spots of TRF1 (Jiang et al., 2007). It would be interesting to see if those larger spots disappear in IE1-expressing cells.

Reference

Jiang W-Q, Zhong Z-H, Henson JD and Reddel RR (2007) Identification of candidate alternative lengthening of telomeres genes by methionine restriction and RNA interference. Oncogene 26: 4635-4647

– The absence of Sp100 does not impact on the formation of the PLM cages but prevent the recruitment of HP1 in those structures. The presence of HP1 in the PML cages could suggest an interaction of HP1 with viral chromatin with nucleosomes containing histone H3 modified on lysine K9 by tri-methylation, for which HP1 is a reader. Could the authors comment on that aspect? Specifically regarding the de novo formation of PML cages around viral genomes by opposition to the relocation of pre-existing PML NBs on incoming viral genomes.

– Although the entrapment of capsids in PML cages is well investigated the question remains to know if the capsids could contain viral genomes or not.

---

## [Author Response]

Reviewer #1 (Recommendations for the authors):The formation of giant PML bodies in response to IE1 deficient HCMV infection is well supported by the compelling microscopy data showing the entrapment of viral genomes within these structures. The aspects related to the importance of viral titer are very interesting, as it seems that these structures are efficiently entrapping viral genomes (as well as viral capsids at different time points) only at the lower MOIs.However, judging from the title, the authors might have aimed at showing that IFN signaling and DNA damage play an important role in formation of these cages, as they can be created in cells even in the absence of viral infection. Although their data do suggest that IFN signaling together with DNA damage response plays a perhaps more important role in the initiation of PML accumulation, these observations were not further exploited and some additional controls could allow for a better understanding of the underlying mechanism.The authors treat HFF with campthotecin or doxyrubicin (inhibitor of TopoI), but do not show any proof of DNA damage. A staining with γH2AX (or TP53BP1) would be helpful to understand if these bodies are formed in the proximity of DNA damage sites.

We have performed the suggested γH2AX staining experiment (see Figure 4i, j): it shows that approximately 80% of all PML ring structures exhibit a colocalization with γH2AX signals after treatment of cells with doxorubicin demonstrating a strong association with DNA damage markers.

Another point that I found would be very interesting to address is the turnover of PML within the CAGE like structures. Are these giant PML bodies susceptible to arsenic trioxide degradation? Arsenic has a well documented role in PML turnover and degradation, and is also used as therapeutic agent in treatment of APL. Moreover, it has also been used in clinical trials for the treatment of HTLV, as well as of HIV-1. It would therefore be very interesting to see if the PML can be degraded by arsenic withing these CAGE like structures, and if this has a relevance for viral reactivation.

We have performed the proposed experiment (see Author response image 1): we found that treatment of HCMVΔhIE1 infected cells with arsenic trioxide induces a general degradation of PML. This affected both PML cages and normal PML-NBs structures. Therefore, be believe that metabolization of PML cages after arsenic trioxide treatment occurs via RNF4. We did not include these data in the manuscript because it does not help to differentiate between effects mediated by PML cages and normal PML-NB structures.

**Author response image 1. sa2fig1:** 

Reviewer #2 (Recommendations for the authors):This reviewer recognises the strength of this submission and its contribution to the field. The points below aim to strengthen the study further to provide greater clarity and context to a non-specialist reader.1. The authors should demonstrate the percentage of viral genomes labelled in their HCMV.EdC virion stock. Based on analogous data derived from HSV-1 (Alandijani et al., 2018), EdC genome labelling efficiency within any given stock of particles can vary. Thus, there may be a significant population of unlabelled HCMV genomes within infected cells that are undetectable by click chemistry. This population should be quantified and presented as supplemental data, as this population of unlabelled genomes may significantly influence the impact and interpretation of data and statements presented in the manuscript. For example lines 271 to 276, 'However, by far not all PML cages contained HCMV genomes suggesting that their enlargement is not a direct consequence of the entrapment process. Taken together, out data provide evidence that genome entrapment by PML-NBs has evolved as a mechanism to achieve efficient and persistent repression of HCMV. Reorganization from dot-like foci to PML cages, however, cannot be directly linked to vDNA entrapment but seems to result from a more general stimulus'.This statement holds true if 90-100% of genomes delivered to the nucleus are detectable by click chemistry. However, if the percentage of genomes labelled is {less than or equal to} 90%, then the authors cannot make this conclusion, as vDNA undetectable by click chemistry may contribute to the phenotypes they observe. Similarly, the authors should make attempts to quantify the precise genome load delivered to cells at the indicated multiplicities of infection used (currently derived from FFU titration). HCMV virus preparations are well established to have a high genome to PFU ratios. Accurate quantitation will provide greater context to non-specialists as to what low or high MOI means by providing absolute numbers for the indicated MOI.

We agree with reviewer #2 that the presence of a significant population of unlabeled viral genomes that are undetectable by click chemistry would affect the interpretation of our results. We tried several times to quantify the labeling efficiency of HCMV genomes as described by *Alandijany et al., 2018,* for herpes simplex virus. However, it turned out that, most probably due to differences in the stability between HCMV and HSV-1 capsids, partial denaturation with 2 M GuHCl does not lead to an ordered release of HCMV vDNA. Thus, we were not able to use this technique for the assessment of vDNA labelling efficiency. Alternatively, we used a method as developed by Christian Sinzger and colleagues (Sinzger et al., J.Gen.Virol. 2007): this method uses pp150 staining as a proxy for capsid localization at the nuclear rim correlating to capsids that have released viral DNA into the nucleus. Pp150 signals located at cell nuclei as well as viral DNA signals in the same nuclei were counted in order to calculate the EdU/EdC labelling efficiency (see Figure 5—figure supplement 2 c, d). From this, we conclude that the EdC labelling efficiency of HCMV DNA is at least 70% (Figure 5—figure supplement 2 d). Although we agree with reviewer #1 that this may lead to a slight underestimation of the number of PML cages associated with vDNA, we still think that our assumption that enlargement is not a direct consequence of the entrapment process holds true due to the facts that vDNA signals were found in both small and large PML foci and, in addition, not all of the large PML cages contained HCMV genomes (see lines 283 – 285).

Furthermore, we agree with reviewer #1 that HCMV virus preparations have a high genome to PFU ratio. This is even aggravated in nucleoside labelling experiments of viral genomes due the necessary purification procedure. Figure 5—figure supplement 2, panel b, shows the quantification of viral genomes in supernatants and infected cells (nuclei and cytoplasm) in relation to immediate-early units (IEUs). According to this quantification, the ratio between intracellular genome equivalents and IEUs was 55. However, the exact quantification of genomes delivered to nuclei is difficult since many genomes as detected by qPCR in infected cells (see Figure 5—figure supplement 2) do not reach the nucleus. By counting vDNA signals of cells infected at an MOI of 0.1 we calculated that this results, on average, in 1.2 viral genomes/cell (Figure 5—figure supplement 2, panel c). However, we anticipate that this value may not only depend on the used virus stock but also on cellular factors (for instance, on the passage number of primary fibroblasts; e.g. HFFs stably expressing individual PML isoforms may vary in their passage number due to the necessary selection procedure). For this reason, we think that accurate quantification is only valid if done for a specific experimental condition.

2. One of the driving conclusions presented by the authors is the requirement for the DDR pathway to be activated to form PML-cages. While the authors present compelling data for this in the context of drug induced DNA damage on uninfected cells (Figure 4g), the authors fail to make the same connection during HCMV infection. The authors should examine the localization of PML-cages induced during deltaIE1-HCMV infection to that of γ-H2Ax, a marker of DNA damage. If their hypothesis holds true, PML-cages should localize in proximity to sites of cellular chromatin damage, thereby providing supporting evidence to one of the papers major conclusions. Alternatively, the authors should investigate the influence of DDR inhibitors (e.g. ATM, ATR, and DNA-PKcs inhibitors) on the frequency PML-cage formation to provide the supporting evidence required to make this conclusion stand during HCMV infection. Correspondingly, if vDNA replication is not required for PML-cage assembly, when is the DDR pathway activated during HCMV infection to promote PML-cage assembly?

Our response:

We have performed the suggested experiments:

– Figures 4i and j show that PML cages co-localize with γH2AX in IFN/Dox-treated non-infected HFFs;

– Figures 5g and h demonstrate an association of entrapped vDNA with DNA damage sites after infection with HCMVΔhIE1;

– Figures 4k and l show that treatment of HCMVΔhIE1-infected cells with the ATM inhibitor Ku55933 results in a statistically significant reduction of enlarged PML cages. Since this could be detected both after high (viral DNA replication is initiated) and low MOI infection (no viral DNA replication occurs), we conclude that incoming viral genomes are sufficient to trigger DNA damage signaling.

3. The manuscript lacks detailed quantitation in its image analysis. While some experiments are quantified, many of the figures (e.g. figures 1 and 2) only show representative images. This makes it hard for the reader to determine the relative population of cells that induce the assembly of PML-cages within any given population of infected cells. Is this phenotype rare or frequent within infected cells? It is recommended that all immunostaining experiments are quantified for readership clarity over a minimum of three independent biological replicates (see below).

We have performed detailed quantifications over at least three independent biological replicates for the experiments of Figures: 1 (see panel b); 3 (see panels b and f); 4 (see panels f, h, j, k and l); 5 (see panels a, d and h); 6 (see panels d and e), Figure 5—figure supplement 2.

Reviewer #3 (Recommendations for the authors):A. Major items to be addressedGeneral:– The reviewer would suggest, to strengthen the physiological relevance of the formation of the PML cages around HCMV genomes and capsids, to perform, if possible, some of the experiments in bone marrow CD34+ cells and/or THP1 cells (already used by the team as HCMV model of latency in previous studies). This, to reviewer's view, could improve significantly, and beyond the molecular aspects, the understanding of the role of the PML NBs in the control of HCMV infection and their role in the establishment of latency and abortive productive infection through the formation of PML cages.

As already stated above, we have previously shown that THP1 cells with an shRNA instituted knockdown of PML do not exhibit a defect concerning the establishment of HCMV latency (Wagenknecht et al., Viruses 2015). Thus, we assume that cellular repressors different from PML-NBs are decisive for HCMV latency establishment.

– l446-448: In the discussion the authors state: "The fact that PML cages form in absence of viral DNA replication implies a formation or incorporation of viral capsids in pre-assembled PML structures rather than an active envelopment of nuclear capsids by PML (Figure 3e, f). » This sentence mingles two aspects of the study : (1) The formation of PML cages around incoming viral genomes; and (2) the formation of PML cages containing viral capsids. To reviewer's opinion the statement, as written, does not consider the phase separation properties of the PML protein (Banani et al., 2016), and for reviews see (Lallemand-Breitenbach and Thé, 2018; Corpet et al., 2020), and the fact that 90% of PML protein is not associated with PML NBs, and has a diffuse nuclear localization (see Lallemand-Breitenbach and Thé, 2010) and/or participating to chromatin dynamics (for a review see Delbarre and Janicki, 2021). The reviewer would suggest to clarify this statement, which is, to reviewer's opinion, misleading on the real capacity of PML NBs to form de novo around genome loci (see Everett and Murray. ND10 Components Relocate to Sites Associated with Herpes Simplex Virus Type 1 Nucleoprotein Complexes during Virus Infection, JVI, 2005, 10.1128/jvi.79.8.5078-5089.2005), and/or viral capsids. May be the authors should consider the most likely possibility that it could exist different mechanisms for the formation of PML cages entrapping incoming viral genomes, and PML cages entrapping capsids.ReferencesBanani SF, Rice AM, Peeples WB, Lin Y, Jain S, Parker R and Rosen MK (2016) Compositional Control of Phase-Separated Cellular Bodies. Cell 166: 651-663Corpet A, Kleijwegt C, Roubille S, Juillard F, Jacquet K, Texier P and LOMONTE P (2020) PML nuclear bodies and chromatin dynamics: catch me if you can! Nucleic Acids Res 389: 251Delbarre E and Janicki SM (2021) Modulation of H3.3 chromatin assembly by PML: A way to regulate epigenetic inheritance. Bioessays: 2100038Lallemand-Breitenbach V and Thé H de (2010) PML nuclear bodies. Cold Spring Harbor perspectives in biology 2: a000661Lallemand-Breitenbach V and Thé H de (2018) PML nuclear bodies: from architecture to function. Curr Opin Cell Biol 52: 154-161

We are well aware of the fact that PML-NBs can form de novo around genome loci (as published by Everett and Murray in 2015). We do not intend to mingle “formation of PML cages around incoming viral genomes” and “formation of PML cages containing viral capsids”. As defined on page 6 of the manuscript the term PML cages is exclusively reserved for enlarged, ring-like PML structures that develop during the late phase of infection. However, we noticed that in some sentences the term PML cages was used, however, instead, PML-NBs were meant. This has been corrected in the revised manuscript. Entrapment of incoming viral genomes as described by Everett and Murray is due to PML-NBs and not due to PML cages. The statement of the Discussion "The fact that PML cages form in absence of viral DNA replication implies a formation or incorporation of viral capsids in pre-assembled PML structures rather than an active envelopment of nuclear capsids by PML” only refers to enlarged PML ring structures surrounding viral capsids. The statement should indicate that PML cages form in the absence of viral capsids. One potential interpretation is that ring-like PML cages are present in a pre-assembled form and subsequently associate with capsids. The statement does not exclude other mechanisms.

Specific– The reviewer acknowledges the importance of IFN signaling and DNA damage for the formation of PML cages, caps or forks in non-infected cells. The requirement of IFN signaling for the formation of PML cages in HCMV-infected cells is clearly shown. However, it is not clear in reviewer's mind how DNA damage provoked by the injection of viral genomes in the nucleus is required to lead to the formation of PML cages specifically around the viral genomes. Does the authors infer that viral genomes themselves would sustain damage that would lead to aggregation of PML around those viral genomes? To reviewer's view the requirement of DNA damage for the formation of PML cages specifically around HCMV genome is not definitively proven. The reviewer would suggest to the authors to perform a γ-H2A.X and PML staining together with click chemistry for the visualization of the genomes at different times post infection to see if PML cages-entrapped viral genomes could at a particular stage of the entry in the nucleus be associated with DNA damage. This could then strengthen the discussion about that aspect.

As already stated in our response to issues raised by reviewers #1 and #2, we have performed co-staining of vDNA with γH2AX at 8 and 72 hpi (see Figure 5g and h of revised manuscript). Furthermore, we show that inhibition of DNA damage signalling by the ATM inhibitor Ku 55933 significantly reduces the formation of PML cages (see Figure 4k and l).

B. OtherSpecific– What is the fate of the capsids that are entrapped in the PML cages? Are they degraded? Do their remain as an abortive, latency-prone, productive infection? Although the reviewer acknowledges the fact that performing such experiments is beyond the scoop of this study, the reviewer would be grateful to the authors to have their feedback on that question. May be this thought could be part of the discussion.

We agree with reviewer #3 that a final answer to this question will require further experimentation. However, since we observed that IE1 expression disrupts PML cages and this correlates with an increase of viral genomes and gene expression, we would assume that PML cage disruption might also result in a release of entrapped capsids (see Figure 6d and e).

– During infection of cells supporting HCMV latency what is the status of IE1 expression? In other words, does it exist a physiological situation of infection by HCMV where IE1 cannot be produced, and consequently cannot induce the dispersal of PML NBs? This should be discussed.

Many authors define HCMV latency as a status of transcriptional silencing of the HCMV MIE locus (e.g. reviewed in Dooley and O’Connor, Pathogens 2020). Consequently, IE1 is not expressed during latency. However, since our paper does not address the role of PML-NBs during latency, we do not want to include this in the Discussion.

– Figure 4g: Could the caps and/or forks in non-infected cells be associated to nucleoli or specific chromosome loci?

Since previous publications reported on PML protein nucleolar associations upon DNA damage induced senescence it is possible that at least some of the enlarged PML structures could be located at nucleoli (Imrichova et al., Aging 2019).

- Figure 6c,d,e: The impact of IE1 expression on the abrogation of HCMV repression leads to the increase of viral genomes and to the expression of at least UL44. IF data to visualize UL44 as a marker of the viral replication compartments and/or MCP for capsids in IE1 expressing cells should be provided in addition to the quantification data.

We have now included IF data to visualize UL44 as a marker of viral replication compartments (see Figure 6c).

– Figure 6 f and g – The low resolution of the pdf combined to the size of theimages make it difficult to conclude anything on the disruption of the APB by IE1. TRF1 is hardly visible in figure 6f. In addition the reviewer would recommend to provide a TRF1 and IE1 staining in figure 6g in addition to the PML/IE1. Indeed, APB are formed by the grouping of several telomeres in the APBs visualized by IF by larger spots of TRF1 (Jiang et al., 2007). It would be interesting to see if those larger spots disappear in IE1-expressing cells.ReferenceJiang W-Q, Zhong Z-H, Henson JD and Reddel RR (2007) Identification of candidate alternative lengthening of telomeres genes by methionine restriction and RNA interference. Oncogene 26: 4635-4647

It is well recognizable in Figure 6g that expression of IE1 induces a complete dispersal of all PML dot-like accumulations. This was true for all cells that were positive for IE1. For that reason, we do not think that a co-staining of TRF1 together with IE1 would give additional insights. We agree that for Figure 6f the signals for TRF1 are hardly visible. Therefore, we exchanged the respective figure (see new Figure 6 f).

– The absence of Sp100 does not impact on the formation of the PLM cages but prevent the recruitment of HP1 in those structures. The presence of HP1 in the PML cages could suggest an interaction of HP1 with viral chromatin with nucleosomes containing histone H3 modified on lysine K9 by tri-methylation, for which HP1 is a reader. Could the authors comment on that aspect? Specifically regarding the de novo formation of PML cages around viral genomes by opposition to the relocation of pre-existing PML NBs on incoming viral genomes.

We would assume that HP1 contributes to HCMV genome repression. Future experiments might verify this by using HP1 knockdown cells in infection experiments. Concerning the question of de novo formation of PML cages versus relocation of pre-exisiting PML-NBs on incoming viral genomes: we already clarified above that Everett and Murray convincingly demonstrated the de novo formation of PML-NBs on incoming viral genomes. PML cages, however, only form late during infection and may represent pre-existing structures to associate with viral capsids.

– Although the entrapment of capsids in PML cages is well investigated the question remains to know if the capsids could contain viral genomes or not.

We agree with reviewer #3 that this is an open question that we plan to address in future experiments.